# MODEL-FREE, REGRET-OPTIMAL BEST POLICY IDENTIFICATION IN ONLINE CMDPS

## ABSTRACT

This paper considers the best policy identification (BPI) problem in online Constrained Markov Decision Processes (CMDPs). We are interested in algorithms that are model-free, have low regret, and identify an optimal policy with a high probability. Existing model-free algorithms for online CMDPs with sublinear regret and constraint violation do not provide any convergence guarantee to an optimal policy and provide only average performance guarantees when a policy is uniformly sampled at random from all previously used policies. In this paper, we develop a new algorithm, named Pruning-Refinement-Identification (PRI), based on a fundamental structural property of CMDPs we discover, called *limited stochasticity*. The property says for a CMDP with $N$ constraints, there exists an optimal policy with *at most $N$* stochastic decisions.

The proposed algorithm first identifies at which step and in which state a stochastic decision has to be taken and then fine-tunes the distributions of these stochastic decisions. PRI achieves trio objectives: (i) PRI is a model-free algorithm; and (ii) it outputs a near-optimal policy with a high probability at the end of learning; and (iii) in the tabular setting, PRI guarantees $\tilde{\mathcal{O}}(\sqrt{K})$[1] regret and constraint violation, which significantly improves the best existing regret bound $\tilde{\mathcal{O}}(K^{\frac{4}{5}})$ under a model-free algorithm, where $K$ is the total number of episodes.

## 1 INTRODUCTION

In unconstrained reinforcement learning (RL), an agent aims to find the optimal policy that maximizes the accumulated reward by interacting with a stochastic environment. RL has achieved remarkable success in multiple areas, including industrial process optimization, robotics, and gaming. (Rajawat et al., 2023; Abeyruwan et al., 2023; Lindegaard et al., 2023; Liu et al., 2023). However, in many real-world applications, the learned policy must also satisfy a set of constraints. For example, in healthcare applications, we need to optimize patient treatment plans while considering constraints like medication dosage, scheduling of medical procedures, and resource allocation in hospitals. These constrained versions of RL problems can be formulated as Constrained Markov Decision Processes (CMDPs) (Altman, 1999).

Learning in CMDPs has become an active research topic recently. Existing solutions include both model-based (Brantley et al., 2020; Efroni et al., 2020; Singh et al., 2020; Liu et al., 2021; Bura et al., 2021; Ding et al., 2021; Chen et al., 2022) and model-free algorithms (Wei et al., 2022b; Ghosh et al., 2022; Wei et al., 2022a). This paper focuses on model-free approaches for CMDPs due to their computation and memory efficiency. A fundamental drawback of existing model-free algorithms for best policy identification in online CMDPs is that they provide only average performance guarantees for a policy uniformly sampled at random from *all* previously used policies during learning, so they fail to identify a single optimal or a near-optimal policy.[2] Therefore, a natural question arises:

**Is it possible to identify an optimal or a near-optimal policy in online CMDPs with the model-free approach with optimal regret?**

---

[1]**Notation:** $f(n) = \tilde{\mathcal{O}}(g(n))$ denotes $f(n) = \mathcal{O}(g(n)\log^k n)$ with $k > 0$. The same applies to $\tilde{\Omega}$.

[2]In this paper, a policy is a mapping from a state at a given step to an action distribution, without any other additional input information. An algorithm that uses multiple policies, e.g. randomly sampling one policy from many policies, is explicitly called a *mixed* policy.

There are two key challenges to answering this question: $(i)$ CMDP problems are typically represented as Linear Programming (LP) problems, resulting in stochastic optimal policies. Model-free online CMDP algorithms often employ the primal-dual approach, utilizing Lagrange multipliers to balance reward maximization and constraint violation. However, these methods yield "greedy policies" for fixed Lagrange multipliers, which aren't necessarily optimal. Consequently, model-free algorithms such as Triple-Q Wei et al. (2021) offer performance guarantees only in terms of averages over various greedy policies determined by different Lagrange multipliers, failing to converge to a single policy. $(ii)$ The best-known regret bound of model-free algorithms for episodic, online CMDPs is $\tilde{\mathcal{O}}(K^{\frac{4}{5}})$ Wei et al. (2022a). It is also known that model-based algorithms can achieve a smaller and order-wise tight regret $\tilde{\mathcal{O}}(\sqrt{K})$ Efroni et al. (2020). The open question is whether a model-free algorithm can reach $\tilde{\mathcal{O}}(\sqrt{K})$ regret in online CMDPs?

This paper tackles both challenges, providing affirmative responses to both questions. We introduce a novel algorithm, Pruning-Refinement-Identification (PRI), rooted in a fundamental CMDP property we unveil—limited stochasticity. This property asserts that for an episodic CMDP with $N$ constraints, there exists an optimal policy making stochastic decisions in at most $N$ step-dependent states out of the $HS$ step-dependent states, where $H$ denotes episode length and $S$ represents possible states at each step.

Based on this insight, PRI consists of three phases. In this first phase (pruning), PRI identifies when and where stochastic decisions are necessary. This defines a set of greedy policies that together approximate a "mixed" optimal policy. The subsequent refinement phase involves learning the weights of these greedy policies. This is done through iterative optimization, utilizing empirical reward and utility value functions. The process refines value function estimates with each iteration, aiming to minimize regret. In the final identification phase, PRI learns the occupancy measure, determining the probability of visiting specific state-action pairs at each step. This information is used to recover a single policy from the near-optimal mixed policy obtain during the refinement phase. The main contributions of this paper include:

- PRI is the first model-free PAC RL algorithm for CMDPs, achieving optimal regret and minimal constraint violation.

- PRI outputs a near-optimal policy with a high probability at the end. The learned policy has $\tilde{\mathcal{O}}(1/\sqrt{K})$ optimality gap with probability $1 - \tilde{\mathcal{O}}(K^{-0.1})$.

- In the tabular setting, PRI guarantees $\tilde{\mathcal{O}}(\sqrt{K})$ regret and constraint violation, which significantly improves the best existing regret bound $\tilde{\mathcal{O}}(K^{\frac{4}{5}})$ under a mode-free algorithm, where $K$ is the total number of episodes. Unlike existing regret bounds, the dominating term in terms of $K$ in the regret bound does not depend on the sizes of the state space and action space.

## 2 RELATED WORK

**Best policy identification in MDPs.** For unconstrained MDPs, existing studies on BPI focus on $(\epsilon, \delta)$-PAC RL algorithms, i.e., algorithms that identify an $\epsilon$-optimal policy with probability at least $1 - \delta$. Such a learning objective has been considered extensively in discounted and episodic tabular MDPs (Agarwal et al., 2020; Azar et al., 2013; Even-Dar et al., 2006; Domingues et al., 2021; He et al., 2021; Sidford et al., 2018). A recent work Taupin et al. (2022) also studied BPI in linear MDPs, which has a sample complexity of $O(\frac{1}{\epsilon^2})$. In other words, the best regret result it might get is $O(\sqrt{K})$. This paper considers BPI for CMDPs using a model-free approach. To the best of our knowledge, it remains an open question.

**Model-based and Model-free algorithms for online CMDPs.** As mentioned in the introduction, most existing results on online CMDPs consider regret minimization. For example, Brantley et al. (2020); Efroni et al. (2020); Singh et al. (2020) proposed model-based algorithms for episodic tabular CMDPs. Liu et al. (2021); Bura et al. (2021) proposed efficient algorithms with zero or bounded constraint violation. For model-free algorithms, Wei et al. (2022b) developed Triple-Q that achieves sublinear regret and zero constraint violation in episodic tabular CMDPs. Similar results have been established for linear CMDPs (Ghosh et al., 2022; Ding et al., 2021) and infinite-horizon average CMDPs (Chen et al., 2022; Wei et al., 2022a). However, these existing model-free algorithms for online CMDPs does not converge to an optimal or a near-optimal policy. Note that model-free

Table 1: The Exploration-Exploitation Tradeoff in Episodic CMDPs.

| | Algorithm | Regret | Constraint Violation | BPI? |
|---|---|---|---|---|
| Model-based | OPDOP (Ding et al., 2021) | $\tilde{\mathcal{O}}(H^3\sqrt{S^2AK})$ | $\tilde{\mathcal{O}}(H^3\sqrt{S^2AK})$ | ✗ |
| | OptDual-CMDP (Efroni et al., 2020) | $\tilde{\mathcal{O}}(H^2\sqrt{S^3AK})$ | $\tilde{\mathcal{O}}(H^2\sqrt{S^3AK})$ | ✗ |
| | OptPrimalDual-CMDP (Efroni et al., 2020) | $\tilde{\mathcal{O}}(H^2\sqrt{S^3AK})$ | $\tilde{\mathcal{O}}(H^2\sqrt{S^3AK})$ | ✗ |
| | CONRL (Brantley et al., 2020) | $\tilde{\mathcal{O}}(H^3\sqrt{S^3A^2K})$ | $\tilde{\mathcal{O}}(H^3\sqrt{S^3A^2K})$ | ✗ |
| | OptPess-LP (Liu et al., 2021) | $\tilde{\mathcal{O}}(H^3\sqrt{S^3AK})$ | 0 | ✗ |
| | OptPess–PrimalDual (Liu et al., 2021) | $\tilde{\mathcal{O}}(H^3\sqrt{S^3AK})$ | $\mathcal{O}(1)$ | ✗ |
| | OPSRL(Bura et al., 2021) | $\mathcal{O}(\sqrt{S^4H^7AK})$ | 0 | ✗ |
| Model-free | Triple-Q(Wei et al., 2022a) | $\tilde{\mathcal{O}}(\frac{1}{\delta}H^4S^{\frac{1}{2}}A^{\frac{1}{2}}K^{\frac{4}{5}})$ | 0 | ✗ |
| | **PRI** | $\tilde{\mathcal{O}}(\sqrt{H^2K})$ | $\tilde{\mathcal{O}}(\sqrt{H^2K})$ | ✓ |

algorithms have a memory complexity of $O(HSA)$ for maintaining the Q-table while the memory complexity of model-based algorithms is $O(HS^2A)$ for maintaining the transition kernel. Very recently, Moskovitz et al. (2023) considered BPI for online CMDPs. They formulated the CMDP problem as a min-max game and the proposed algorithm converges to a near-optimal policy at the last iteration with optimistic mirror descent. However, the paper does not provide any regret guarantee when learning the near-optimal policy. There are also algorithms for the average-reward CMDP problem, including model-based approaches Agarwal et al. (2021; 2022); Zheng & Ratliff (2020) and model-free approaches Chen et al. (2022); Wei et al. (2022b). These algorithms do not identify the optimal policy at the end. Table 1 summarizes the recent results on online, episodic CMDPs.

## 3 PROBLEM FORMULATION

We consider an episodic CMDP, denoted by $(\mathcal{S}, \mathcal{A}, H, \mathbb{P}, r, g^n, n \in [N])$, where $\mathcal{S}$ is the state space ($|\mathcal{S}| = S$), $\mathcal{A}$ is the action space ($|\mathcal{A}| = A$), $\{r_h\}_{h=1}^H, \{g_h^n\}_{h=1}^H, n \in [N]$ are reward, $n$-th utility functions, and $\mathbb{P} = \{\mathbb{P}_h(\cdot|x,a)\}_{h=1}^H$ are the transition kernels. For simplicity, we assume that in each episode, the agent starts from the same initial state $x_1 = x_{ini}$. It is straightforward to generalize the results to the case when the initial state is sampled with a given distribution but the notation becomes cumbersome. We also assume that $r_h : \mathcal{S} \times \mathcal{A} \to [0,1]$ and $g_h^n : \mathcal{S} \times \mathcal{A} \to [0,1]$ are deterministic for notation simplicity. Our results can be easily generalized to random reward/utility signals.

During each episode, the agent interacts with the environment as follows: at each step $h$, the agent takes action $a_h$ after observing state $x_h$, receives reward $r_h(x_h, a_h)$ and $N$ utility values $g_h^n(x_h, a_h)$ ($n \in [N]$), and then observes a new state ($x_{h+1}$), which evolves by following the transition kernel $\mathbb{P}_h(\cdot|x_h, a_h)$. The episode terminates after $H$ steps.

Given a stochastic policy $\pi$, which is a collection of $H$ functions $\{\pi_h : \mathcal{S} \times \mathcal{A} \to [0,1]\}_{h=1}^H$, the agent takes action $a$ with probability $\pi_h(a|x)$ when being in state $x$ at step $h$ . The reward value function of policy $\pi$, denoted by $V_h^\pi(x)$, is the expected total reward when starting from an arbitrary state $x$ at step $h$ to the end of the episode: $V_h^\pi(x) = \mathbb{E}_\pi\left[\sum_{i=h}^H r_i(x_i, a_i)\Big| x_h = x\right]$, where the expectation is taken with respect to the policy $\pi$ and randomness from the transition kernels. Accordingly, the reward Q-function, denoted by $Q_h^\pi(x,a)$, is the expected total reward when the agent starts from an arbitrary action-action pair $(x, a)$ at step $h$ and follows policy $\pi$ to the end of the episode: $Q_h^\pi(x,a) = r_h(x,a) + \mathbb{E}_\pi\left[\sum_{i=h+1}^H r_i(x_i, a_i)\Big| x_h = x, a_h = a\right]$.

Similarly, we can define the $N$ utility value functions as $W_h^{\pi,n}(x) = \mathbb{E}_\pi\left[\sum_{i=h}^H g_i^n(x_i, a_i)\Big| x_h = x\right]$ and utility Q-functions as $C_h^{\pi,n}(x,a) = g_h^n(x,a) + \mathbb{E}_\pi\left[\sum_{i=h+1}^H g_i^n(x_i, a_i)\Big| x_h = x, a_h = a\right]$. Given the definitions above, we have

$$V_h^\pi(x) = \sum_a \pi_h(a|x)Q_h^\pi(x,a) \quad Q_h^\pi(x,a) = r_h(x,a) + \sum_{x'} \mathbb{P}_h(x'|x,a)V_{h+1}^\pi(x') \tag{1}$$

$$W_h^{\pi,n}(x) = \sum_a \pi_h(a|x)C_h^{\pi,n}(x,a) \quad C_h^{\pi,n}(x,a) = g_h^n(x,a) + \sum_{x'} \mathbb{P}_h(x'|x,a)W_{h+1}^{\pi,n}(x'). \tag{2}$$

The objective of the CMDP is to find an optimal policy that maximizes the expected total reward while making sure the $n$−th expected total utility is no less than $\rho^{(n)}$ for all $n \in [N]$:

$$\pi^* \in \arg\max_\pi V_1^\pi(x_{ini}) \quad \text{s.t.} \quad W_1^{\pi,n}(x_{ini}) \geq \rho^{(n)} \quad \forall n \in [N]. \tag{3}$$

To avoid triviality, we assume $\rho^{(n)} \in [0, H]$. For simplicity, we use $V_1^\pi$ to represent $V_1^\pi(x_{ini})$ and $W_1^{\pi,n}$ to represent $W_1^{\pi,n}(x_{ini})$.

We evaluate an online RL algorithm for CMDP using regret and constraint violation over $K$ episodes:

$$\text{Regret}(K) = K V_1^{\pi^*}(x_{ini}) - \mathbb{E}\left[\sum_{k=1}^K V_1^{\pi_k}(x_{ini})\right] \tag{4}$$

$$\text{Violation}^n(K) = K\rho^{(n)} - \mathbb{E}\left[\sum_{k=1}^K W_1^{\pi_k,n}(x_{ini})\right] \tag{5}$$

where $\pi_k$ is the policy used in episode $k$.

## 4 PRI (PRUNING-REFINEMENT-IDENTIFICATION)

Before formally introducing our algorithm, we first present two structural properties of the optimal solution to the CMDP problem equation 3. These properties have been overlooked in the literature but serve as the foundation of our proposed algorithm. Consider a CMDP problem with $N$ constraints. It is well-known that the problem can be formulated as a linear programming (LP) problem Altman (1999):

$$\max_{\{q_h(x,a)\}} \sum_{h,x,a} q_h(x,a)r_h(x,a) \tag{6}$$

$$\text{s.t.:} \sum_{h,x,a} q_h(x,a)g_h^{(n)}(x,a) \geq \rho^{(n)} \quad \forall n \in [N] \tag{7}$$

$$\sum_a q_{h+1}(x,a) = \sum_{x',a'} \mathbb{P}_h(x|x',a')q_h(x',a') \quad \forall x \in \mathcal{S}, h \in [H] \tag{8}$$

$$\sum_a q_1(x_{ini},a) = 1, \sum_a q_1(x,a) = 0, \ x \neq x_{ini} \tag{9}$$

$$q_h(x,a) \geq 0, \tag{10}$$

where $q_h(x,a)$ denotes the probability that state-action pair $(x,a)$ is visited at step $h$, called the occupancy measure. Each feasible solution $\{q_h(x,a)\}_{h,x,a}$ to the problem leads to a corresponding Markov policy: $\pi_h(a|x) = \frac{q_h(x,a)}{\sum_a q_h(x,a)}$. In this paper, we call probability distribution $\pi_h(\cdot|x)$ *decision* at state $x$ at step $h$. So a policy consists of $S \times H$ decisions. A decision $\pi_h(\cdot|x)$ is called *greedy* if $\pi_h(a|x) = 1$ for some $a \in \mathcal{A}$ and stochastic otherwise.

**Lemma 1** (Limited Stochasticity). *If $q^* = \{q_h^*(x,a)\}_{h,x,a}$ is an optimal solution to the CMDP problem equation 6-equation 10 and is an extreme point, then there are at most $HS + N$ nonzero values in $q^*$. This implies that the optimal policy derived from $q^*$ includes at most $N$ stochastic decisions.*

The detailed proof can be found in Appendix B. The following corollary, which is a well-known result, is a direct consequence of the lemma.

**Corollary 1.** *For unconstrained MDP problems, one of the optimal policies is a greedy policy.*

Given an occupancy measure $q$ and its induced policy $\pi$, we define $\mathcal{D}_{h,x}(q) = \{a : q_h(x,a) > 0\}$, which is the set of actions that will be taken with a nonzero probability in state $x$ at step $h$ under the policy $\pi$ induced by $q$. Note that if $\pi_h(\cdot|x)$ is a greedy decision, then $|\mathcal{D}_{h,x}(q)| = 1$; and if $\pi(\cdot|x)$ is greedy, then $|\mathcal{D}_{h,x}(q)| > 1$. Let $M = \prod_{h,x} |\mathcal{D}_{h,x}(q)|$, and let $\pi^m$ represent the $m$th greedy policy $(m = 1, \cdots, M)$ constructed from $\otimes_{h,x}\mathcal{D}_{h,x}(q)$ such that $\pi_h^m(a|x) = 1$ only if $a \in \mathcal{D}_{h,x}(q)$. Note that according to lemma 1, we have $M \leq 2^N$. A greedy policy is a policy under which all decisions are greedy. Next, we will show that a Markov policy is equivalent to a mixed policy of many greedy policies in the following lemma, whose proof can be found in Appendix B.2.

**Lemma 2** (Decomposition). *Given any Markov policy $\pi$ and its corresponding occupancy measure $q$, there exists a set of $M$ greedy policies and a probability distribution $\{a_m\}_{m=1,\cdots,M}$ such that the mixed policy, which selects a greedy policy $\pi^m$ at the start of an episode with probability $a_m$ and subsequently follows it, has the same occupancy measure $q$ as the original policy $\pi$.*

Online model-free algorithms for CMDPs, such as Triple-Q Wei et al. (2021), guarantee sublinear regret and constraint violation on average but have no convergence guarantee. In fact, Triple-Q continues to adjust the dual variable (virtual queue) based on constraint violation, and when the dual variable is fixed (within a frame), the algorithm reduces to the traditional Q-learning. As suggested in the paper Wei et al. (2021), we can only recover a near-optimal policy by remembering all previous policies and then uniformly sampling one from them for each episode, i.e., a mixed policy of $K$ policies. Therefore, this near-optimal policy is a mixture of many, many greedy policies. More importantly, it is near-optimal only when averaging over a large number of episodes and may be far from optimal in each episode.

Hence, unlike unconstrained MDPs where Q-learning converges to the optimal policy, finding a model-free algorithm that converges to the optimal policy or a near-optimal policy in CMDPs is highly nontrivial and remains to be an open problem. Lemma 1 (limited stochasticity), however, suggests that when the number of constraints, $N$, is relatively small, solving an unknown CMDP may not differ significantly from solving an unknown MDP. This is because the majority of the decisions, specifically $HS - N$ out of the $HS$ decisions, are greedy and can be learned using traditional algorithms like Q-learning if we can first identify where the stochastic decisions need to be taken. Lemma 2 further suggests that an optimal policy can be decomposed into $M$ greedy policies if all decision types are correctly identified, so we may recover an optimal or a near-optimal policy by evaluating the $M$ greedy policies.

We will first consider the case where *the LP has a unique optimal solution*. Leveraging these two observations from Lemma 1 and 2, we propose a novel three-phase algorithm (Algorithm 1), including policy pruning, policy refinement, and policy identification, called PRI.

The algorithm is presented in Algorithm 1, which includes $\sqrt{K} + 2K$ episodes, $\sqrt{K}$ episodes for pruning, $K$ episodes for refinement and $K$ episodes for identification. In the first phase (policy pruning), we run Triple-Q for $\sqrt{K}$ episodes, we denote $\{\pi_{k,h}\}_{h=1}^H$ as the policy used by Triple-Q in the $k$th episode, and it is a greedy policy.

**Remark:** For fixed $(h, x, a)$ in the policy pruning phase, we use $\tilde{N}_h(x, a)$ to count the number of episodes in which the greedy policy we follow is $\pi_h(a|x) = 1$, which is the number of greedy policies (among the $\sqrt{K}$ greedy policies) that would take action $a$ if the agent visits state $x$ at step $h$.

Because of the sub-linear regret and zero violation guaranteed by Triple-Q, we expect that $\frac{N_h(x,a)}{\sqrt{K}}$ is close to zero if $\pi_h^*(a|x) = 0$ and is a non-negligible positive value if otherwise. Therefore, with a high probability, $\tilde{\mathcal{D}}_{h,x} = \mathcal{D}_{h,x}(q^*)$, where $\tilde{\mathcal{D}}_{h,x}$ is gradually updated in Algorithm 1 (Lines 8-10).

After the first phase, PRI obtains $M$ greedy policies. In the second phase, PRI learns the weights $\{\alpha_m\}$ so that a mixed policy that chooses policy $\pi^m$ with probability $\alpha_m$ is statistically identical to the optimal policy. This is achieved by learning the reward and utility value functions of the greedy policies and then solving an approximated version of the CMDP (Decomposition-Opt equation 11).

At each round of the second phase (policy refinement), the following optimization with $M$ optimization variables is solved.

$$\textbf{Decomposition-Opt: } \max_{\{a_m\}_{m=1}^M} \sum_{m=1}^M \alpha_m \bar{V}_1^{\pi^m}$$

$$\text{s.t.: } \left| \sum_{m=1}^M \alpha_m \bar{W}_1^{\pi^m,n} - \rho^{(n)} \right| \leq \sqrt{\frac{H^2 \log\left((t-1)\epsilon'K\right)}{\epsilon'(t-1)\sqrt{K}}} \quad \forall n, \qquad (11)$$

$$\sum_m \alpha_m = 1, \alpha_m \geq \epsilon' \quad \forall m.$$

---

**Algorithm 1** PRI

---

1: Phase 1: Policy Pruning
2: Initialize $\tilde{N}_h(x,a) = 0$ for all $h$, $x$ and $a$. Other initialization is the same as in Triple-Q.
3: **for** $k = 1, \cdots, \sqrt{K}$ **do**
4:     For all $(h,x,a)$, $\tilde{N}_h(x,a) \leftarrow \tilde{N}_h(x,a) + \pi_{k,h}(a|x)$.
5:     Execute Triple-Q for one episode.
6: **for** all $(h,x)$ **do**
7:     Initialize $\tilde{\mathcal{D}}_{h,x} = \emptyset$
8:     **for** all $a \in \mathcal{A}$ **do**
9:         $\tilde{\mathcal{D}}_{h,x} \leftarrow \tilde{\mathcal{D}}_{h,x} \bigcup \{a\}$ if $\frac{\tilde{N}_h(x,a)}{\sqrt{K}} \geq \frac{\epsilon}{2}$.
10: Obtain $M$ greedy policies from $\{\tilde{\mathcal{D}}_{h,x}\}_{h,x}$ where $M = \prod_{h,x} |\tilde{\mathcal{D}}_{h,x}|$.
11: Phase 2: Policy Refinement
12: **if** $M = 1$ **then**
13:     Output the greedy policy.
14: **else**
15:     Set $\hat{V}_1^{\pi^m} = 0, \hat{W}_1^{\pi^m,n} = 0$, and $a_m = \frac{1}{M}$ for all $n$ and $m$.
16:     **for** round $t = 1, \cdots, \sqrt{K}$ **do**
17:         **for** $m = 1, \cdots, M$ **do**
18:             **for** $k = 1, \cdots, \alpha_m \sqrt{K}$ **do**
19:                 Execute greedy policy $\pi^m$ for one episode.
20:                 **if** $k \leq \epsilon' \sqrt{K}$ **then**
21:                     Set $\hat{V}_1^{\pi^m} \leftarrow \hat{V}_1^{\pi^m} + V_{k,1}^{\pi^m}$ and $\hat{W}_1^{\pi^m,n} \leftarrow \hat{W}_1^{\pi^m,n} + W_{k,1}^{\pi^m,n}$ for all $n$, where
$V_{k,1}^{\pi^m}$ and $W_{k,1}^{\pi^m,n}$ are the total reward and utility of type $n$ received in the $k$th episode.
22:                     Set $\bar{V}_1^{\pi^m} = \frac{\hat{V}_1^{\pi^m}}{t\epsilon' \sqrt{K}}$ and $\bar{W}_1^{\pi^m,n} = \frac{\hat{W}_1^{\pi^m,n}}{t\epsilon' \sqrt{K}}$ for all $n$.
23:         Update $\{\alpha_m\}$ by solving Decomposition-Opt equation 11.
24: Phase 3: Policy Identification
25: Initialize $N_h(x,a) = 0$ for all $h$, $x$ and $a$.
26: **for** $t = 1, \cdots, \sqrt{K}$ **do**
27:     **for** $m = 1, \cdots, M$ **do**
28:         **for** $k = 1, \cdots, \alpha_m \sqrt{K}$ **do**
29:             **for** $h = 1, \cdots, H$ **do**
30:                 Take action $a_h$ given by policy $\pi^m$, i.e. $\pi^m(a_h|x_h) = 1$.
31:                 $N_h(x_h, a_h) \leftarrow N_h(x_h, a_h) + 1$.
32: For all $(h,x,a)$, set $\tilde{\pi}_h(a|x) = \frac{N_h(x,a)}{\sum_{\bar{a} \in \mathcal{A}} N_h(\bar{a},x)}$.
33: Output policy $\tilde{\pi}$.

---

After learning sufficiently accurate $\{\alpha_m\}$ in the second phase, PRI learns the occupancy measure under the mixed policy defined by $\{\alpha_m\}$ and constructs a Markov policy $\tilde{\pi}$ based on the learned occupancy measure.

In the next section, we will show that PRI guarantees $\mathcal{O}(\sqrt{K})$ regret and constraint violation and outputs a near-optimal policy $\tilde{\pi}$ with a high probability. An informal statement of the main results is presented below. The formal statements of the theorems and the proofs will be presented in the next section.

**Main Results:** Assume the LP defined by equation 6-equation 10 has a unique solution. With a high probability, PRI yields policy $\tilde{\pi}$ such that

- $\{(h,x,a) : \tilde{\pi}_h(a|x) > 0\} = \{(h,x,a) : \pi_h^*(a|x) > 0\}$,

- PRI guarantees $\mathcal{O}(\sqrt{K})$ regret and constraint violation over the $\sqrt{K} + 2K$. episodes, and

- $|\tilde{\pi}_h(a|x) - \pi_h^*(a|x)| = \mathcal{O}(1/\sqrt{K})$ for all $(h,x,a)$, and $\tilde{\pi}_h(a|x) = \pi_h^*(a|x)$ if $\pi_h^*(a|x) \in \{0,1\}$.

**Remark:** If the LP has more than one solution, we will introduce a multi-solution pruning algorithm to the policy pruning phase of PRI to resolve the issue. The algorithm and the analysis can be found in Section 6.

## 5 MAIN RESULTS

In this section, we provide our main results assuming that the LP associated with the CMDP problem has a unique solution. This assumption can be relaxed and the results can be found in Section 6. Let $\pi^*$ be the unique optimal policy and $\{q_h^{\pi^*}(x,a)\}$ is the corresponding occupancy measure. Furthermore, let $\{\pi^m\}$ $(m = 1, \cdots, M)$ be the set of greedy policies associated with the optimal policy as defined in Lemma 2, and $\{a_m^*\}$ the associated weights. We also make the following additional assumptions.

**Assumption 1.** *The $\epsilon$ and $\epsilon'$ used in PRI satisfy $q_h^{\pi^*}(x,a) \geq \epsilon$ for any $(h,x,a)$ such that $\pi_h^*(a|x) > 0$, and $\min_m a_m^* \geq \epsilon' > 0$.*

**Assumption 2.** *There exist two positive constants $c_v$ and $c_w$ such that given a feasible occupancy measure $q^\pi$ to the LP and the corresponding reward value function and utility value function $V^\pi$ and $W^{\pi,n}$, we have either $V_1^{\pi^*} - V_1^\pi \geq c_v \|q^{\pi^*} - q^\pi\|_1$ or for some $n \in [N], W_1^{\pi^*,n} - W_1^{\pi,n} \geq c_w \|q^{\pi^*} - q^\pi\|_1$, where $\|\cdot\|_1$ is the L1-norm.*

Recall a feasible occupancy measure defines a unique Markov policy. The assumption above states that when a policy's occupancy measure is different from that of the unique optimal policy, then either the reward value function or one of the utility reward functions should also be different from that under the optimal policy.

**Assumption 3.** *Under any greedy policy $\pi$, for all $x$ and $h$, we have $\Pr(x_h = x) = \sum_{x',a'} q_{h-1}^\pi(x',a') \mathbb{P}_h(x|x',a') > p_{\min}$.*

This assumption above says all states should be visited with a non-negligible probability under any greedy policy. It is worth noting that this assumption can be removed if we apply the extension version of PRI, which is stated in section 6. To prove our main result, we first recall the regret and constraint violation guaranteed under Triple-Q Wei et al. (2022a) in the following lemma.

**Lemma 3.** *For sufficiently large $K$, over $K$ episodes, Triple-Q guarantees $\tilde{\mathcal{O}}(K^{0.8})$ regret and zero constraint violation, and furthermore,*

$$\Pr\left(K\rho^{(n)} - \sum_{k=1}^K W_1^{\pi_k,n} \leq 0\right) = 1 - \mathcal{O}\left(\frac{1}{K^2}\right). \tag{12}$$

Noting that we use Triple-Q in the pruning phase, the assumptions for Triple-Q like Slater's condition are still required. A brief review can be found in the Appendix A. We remark that PRI can be viewed as a "meta-algorithm" that builds on any model-free CMDP algorithm with sublinear regret and constraint violation. In the following theorem, we show that PRI can correctly classify stochastic and greedy decisions with a high probability after the pruning phase.

**Theorem 1** (Pruning). *Let $\mathcal{D}^* = \{(h,x,a) : \pi_h^*(a|x) > 0\}$ and $\tilde{\mathcal{D}} = \left\{(h,x,a) : \frac{\tilde{N}_h(x,a)}{\sqrt{K}} \geq \frac{\epsilon}{2}\right\}$. Under Assumptions 1 and 3, after policy pruning, we have*

$$\Pr\left(\tilde{\mathcal{D}}_{h,x} = \mathcal{D}_{h,x}(q^*), \forall(h,x)\right) = 1 - \tilde{\mathcal{O}}\left(K^{-0.1}\right). \tag{13}$$

The detailed proof is deferred to Appendix C. Note that since the pruning phase includes $\sqrt{K}$ episodes, the regret and constraint violation are both bounded by $H\sqrt{K}$.

The following theorem shows that the regret and constraint violation during the refinement phase are both $\tilde{\mathcal{O}}(\sqrt{K})$. Note that $\tilde{\mathcal{O}}(\sqrt{K})$ regret and constraint violation imply that the learned mixed policy is close to optimal. The proof can be found in Appendix D.

**Theorem 2** (Refinement). *Assume $\tilde{\mathcal{D}} = \mathcal{D}^*$ after policy pruning. Under Assumption 1 to 3, with probability $1 - \tilde{\mathcal{O}}(\frac{1}{\sqrt{K}})$, the regret and constraint violation during the policy refinement phase are both $\tilde{\mathcal{O}}(H\sqrt{K})$.*

The refinement phase learns a near optimal mixed policy, which is a combination of $M$ greedy policies for $M \leq 2^N$. In the following theorem we show that the identification phase is to find a single near-optimal policy by using the occupancy measure of the mixed policy. The proof can be found in Appendix E.

**Theorem 3** (Identification). *Assume $\tilde{\mathcal{D}} = \mathcal{D}^*$ after policy pruning. Under Assumption 1 to 3, with probability $1 - \tilde{\mathcal{O}}(\frac{1}{K})$, the regret and constraint violation during the policy identification phase are both $\mathcal{O}(\sqrt{K})$. Furthermore, $|\tilde{\pi}_h(a|x) - \pi_h^*(a|x)| = \mathcal{O}(\frac{1}{\sqrt{K}})$ if $0 < \pi_h^*(a|x) < 1$ and $\tilde{\pi}_h(a|x) = \pi_h^*(a|x)|$ if $\pi_h^*(a|x) \in \{0, 1\}$.*

By summarizing the results from the three theorems above, we have the regret and the constraint violation over the $\sqrt{K} + 2K$ episodes are $\tilde{\mathcal{O}}(H\sqrt{K})$ with probability $1 - \tilde{\mathcal{O}}(\frac{1}{K^{0.1}})$. Consider the regret, the pruning phase includes $\sqrt{K}$ episodes, resulting in at most $H\sqrt{K}$ regret. Theorem 2 and Theorem 3 show that the regret in the refinement and identification phases are both $\tilde{\mathcal{O}}(H\sqrt{K})$. Note that the order-wise bounds are independent of $S$ and $A$, unlike those in the literature. However, there is an implicit dependence on $S$ and $A$ as the results hold only when $K$ is sufficiently large and how large $K$ needs to be depends on $S$ and $A$.

## 6 EXTENSION TO CMDPS WITH MULTIPLE OPTIMAL SOLUTIONS

In this section, we consider the case where the optimal policy is not unique so the LP has multiple optimal solutions. Here, the RL agent's objective is to learn one of these optimal policies. According to Lemma 1, an optimal solution associated with an extreme point of the LP involves no more than $HS + N$ stochastic decisions. Additionally, any optimal policy can be viewed as a combination of the optimal policies associated with the extreme points. We define the set of optimal policies as $\Pi^*$ and the subset associated with extreme points as $\Pi^{*,e}$. We expand our assumptions to the case of multiple solutions as follows.

**Assumption 4.** *The $\epsilon$ used PRI satisfies $\min_{(h,x,a):\pi_h(a|x) \neq 0} q_h^\pi(x, a) \geq \epsilon \quad \forall \pi \in \Pi^{*,e}$.*

**Assumption 5.** *Given any occupancy measure $q'$ and the induced Markov policy $\pi'$, there exists an optimal policy $\pi^* \in \Pi^*$ such that $V_1^{\pi^*} - V_1^{\pi'} \geq c_v ||q^{\pi^*} - q'||_1$ or for some $n$, $W_1^{\pi^*,n} - W_1^{\pi',n} \geq c_w ||q^{\pi^*} - q'||_1$, where $c_v$ and $c_w$ are two positive constants.*

Note that if $\pi'$ is an optimal policy, then the assumption holds trivially with $\pi^* = \pi'$. Under Assumptions 3-5, the following theorem shows that a unique optimal policy is identified after Multi-Solution Pruning. The algorithm result in at most $H^2 SAK^{0.25}$ regret and constraint violation.

**Theorem 4.** *Under Assumption 4 and 5, with probability $1 - \mathcal{O}(1/K^{0.02})$, for sufficiently large $K$, multi-solution pruning outputs a unique optimal policy with at most $N$ stochastic decisions. The regret and constraint violation during multi-solution pruning are bounded by $H^2 SAK^{0.25}$ with probability one.*

More discussions and the detailed proof are deferred to Appendix F.1 due to the page limit. We note that adding this multi-solution pruning to PRI only increases the regret and constraint violation by $HSAK^{0.25}$ which is order-wise smaller than $\tilde{\mathcal{O}}(H\sqrt{K})$ in terms of $K$. Therefore, the regret and constraint violation remains to be $\tilde{\mathcal{O}}(H\sqrt{K})$ for sufficiently large $K$. The learned policy is a near optimal policy with $\tilde{O}(1/\sqrt{K})$ gap with probability $1 - \tilde{O}(K^{-0.02})$.

## 7 EXPERIMENTS

**Synthetic CMDP**

This section presents numerical evaluations of the proposed algorithm. We first evaluated our algorithm for a synthetic CMDP with a single constraint. The transition kernels, rewards, and utilities are chosen such that the problem has a unique optimal solution and satisfies Assumption 3. The objective is to maximize the cumulative reward while guaranteeing that the cumulative utility is at least 2. Comparison between Triple-Q and PRI can be found in Figure 1. Experiment details can be found in the Appendix H.1.

We can observe that PRI converges significantly faster than Triple-Q. Remarkably, both regret and constraint violation level off at the beginning of policy refinement after approximately $110,000$ episodes. However, the regret of Triple-Q continues to increase sublinearly. PRI significantly outperforms Triple-Q on regret. At the end of the $1,100,000$ episodes, Triple-Q has a regret of $2.05 \times 10^5$ and constraint violation of $-3.86 \times 10^4$. In contrast, the regret and constraint violation under PRI are $-1.73 \times 10^3$ and $1.06 \times 10^4$, respectively. Thus, the regret is significantly lower than Triple-Q. Since the full CMDP model is given, we can obtain the optimal solution by using the linear programming approach. The cumulative reward and cumulative utility we get for our learned policy are $1.57301$, and $2.00008$, which are very close to the optimal solution $1.57306$ and $2$.

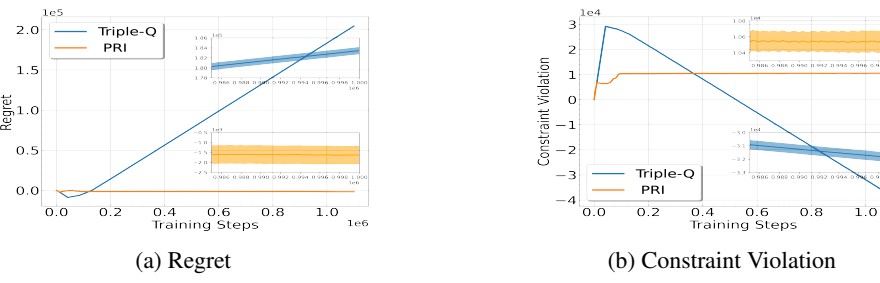

(a) Regret                    (b) Constraint Violation

Figure 1: Results for a synthetic CMDP with a unique solution, the shaded region represents the 95% confidence interval.

**Grid-world**

In our second experiment, which is a grid-world environment (refer to Appendix H.2 for details), we compared Triple-Q with PRI, and the results are shown in Figure 2. This problem has multiple optimal policies. Therefore, we used the extended PRI with multi-solution-pruning. PRI consists of $200,000$ episodes for the initial phase, followed by $200,000$ episodes for each multi-solution pruning phase. Both policy refinement and policy identification phases include $5,000,000$ episodes each. For reference, we ran Triple-Q for the same number of episodes. The outcomes concerning regret and constraint violation are visualized in Figure 2a and 2b. We can observe that Triple-Q has a regret of $3.19 \times 10^6$ and a constraint violation of $-5.26 \times 10^5$, whereas PRI achieves $1.54 \times 10^5$ regret and $2.98 \times 10^3$ constraint violation, indicating substantially lower regret with PRI.

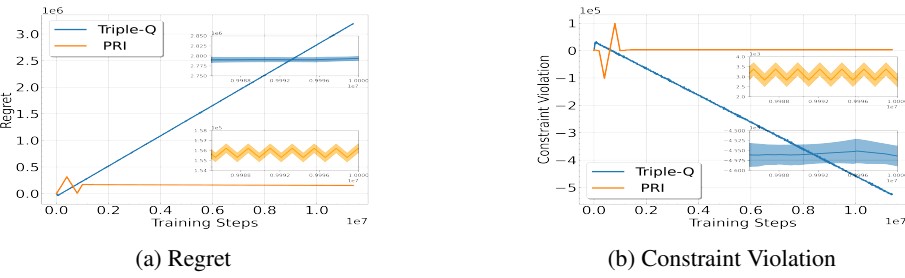

(a) Regret                    (b) Constraint Violation

Figure 2: Results for the grid world environment, the shaded region represents the 95% confidence interval.

## 8    CONCLUSIONS

In this paper, we developed a model-free, regret-optimal algorithm for online CMDPs, called PRI. The algorithm is based on a fundamental observation that for a CMDP with $N$ constraints, there exists an optimal policy that includes at most $N$ stochastic decisions. In the tabular setting, PRI guarantees $\tilde{\mathcal{O}}(\sqrt{K})$ regret and constraint violation and the bounds are independent of the size of state and action spaces for sufficiently large $K$. The same result holds when the violation cannot be canceled across episodes with minor modifications of the algorithm. The details can be found in Appendix G.

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
