# A   REVIEW OF TRIPLE-Q

In this section, we briefly review Triple-Q for CMDPs. The design of Triple-Q is based on the primal-dual approach in optimization. The notions used in this section may be slightly abused, but we will make sure the definitions are clear. Consider the case with only one constraint for simplification. Given a Lagrange multiplier $\lambda$, we consider the Lagrangian of the problem from a given initial state $x_1$ :

$$\max_{\pi} V_1^{\pi}(x_1) + \lambda \left( W_1^{\pi}(x_1) - \rho \right)$$

$$= \max_{\pi} \mathbb{E}\left[ \sum_{h=1}^{H} r_h(x_h, \pi_h(x_h)) + \lambda g_h(x_h, \pi_h(x_h)) \right] - \lambda \rho,$$

which is an unconstrained MDP with reward $r_h(x_h, \pi_h(x_h)) + \lambda g_h(x_h, \pi_h(x_h))$ at step $h$. Assuming we solve the unconstrained MDP and obtain the optimal policy, denoted by $\pi_{\lambda}^*$, we can then update the dual variable (the Lagrange multiplier) using a gradient method:

$$\lambda \leftarrow \left( \lambda + \rho - \mathbb{E}\left[ W_1^{\pi_{\lambda}^*}(x_1) \right] \right)^+. \tag{14}$$

While primal-dual is a standard approach, analyzing the finite-time performance, such as regret or sample complexity, is particularly challenging. Triple is designed as a two-time scale algorithm for addressing the trade-off between regret and constraint violations.

- At each step, Triple-Q updates two the Q-values for $(x_{h-1}, a_{h-1})$ after observing $(s_h, a_h)$, reward $r_h(x_h, a_h)$ and utility $g_h(x_h, a_h)$ in a fast time scale. In particular,

$$Q_{h-1}(x_{h-1}, a_{h-1}) \leftarrow (1 - \alpha_t)Q_{h-1}(x_{h-1}, a_{h-1}) + \alpha_t \left( r_{h-1}(x_{h-1}, a_{h-1}) + V_h(x_h) + b_t \right)$$
$$C_{h-1}(x_{h-1}, a_{h-1}) \leftarrow (1 - \alpha_t)C_{h-1}(x_{h-1}, a_{h-1}) + \alpha_t \left( g_{h-1}(x_{h-1}, a_{h-1}) + W_h(x_h) + b_t \right)$$

- at the end of each frame, the virtual queue length is updated in a slow time scale manner as $\left( Z + \rho + \epsilon - \frac{\bar{C}}{K^{\alpha}} \right)^+$, where $\bar{C}$ is the average of all the Q-values of the utility function at the initial state-action $(x_1, a_1)$.

The algorithm only needs to know the values of $H$, $A$, $S$ and $K$, and no other problem-specific values are needed. Furthermore, Triple-Q includes updates of two Q-functions per step: one for $Q_h$ and one for $C_h$; and one simple virtual queue update per frame. So its computational complexity is similar to SARSA.

# B   PROOFS OF THE TECHNICAL LEMMAS

## B.1   PROOF OF LEMMA 1 (LIMITED STOCHASTICITY)

**Lemma 1.** *If $q^* = \{q_h^*(x, a)\}_{h,x,a}$ is an optimal solution to the CMDP problem equation 6-equation 10 and is an extreme point, then there are at most $HS + N$ nonzero values in $q^*$. This implies that the optimal policy derived from $q^*$ includes at most $N$ stochastic decisions.*

*Proof.* The LP has $HSA$ decision variables $\{q_h(s, a)\}$ in total. So at an extreme point, at least $HSA$ constraints become tight. In other words, at least $HSA$ constraints become equalities under solution $q^*$. Since there are only $HS + N$ constraints defined in equation 7-equation 9, at least

$$HSA - HS - N = HS(A - 1) - N$$

constraints in equation 10 become tight (equality) under $q^*$. Therefore, there are at least $HS(A - 1) - N$ zeros in $q^*$ or at most $HS + N$ nonzero values in $q^*$.

Now suppose the optimal policy obtained from $q^*$ has less than $HS - N$ greedy decisions. Then $q^*$ would have at least

$$HS - N - 1 + 2(N + 1) = HS + N + 1$$

nonzero values because each greedy decision requires one nonzero $q_h(x, a)$ and each stochastic decision requires at least two nonzero $q_h(x, a)$. This leads to a contradiction. $\square$

### B.2 PROOF OF LEMMA 2 (DECOMPOSITION)

**Lemma 2.** *Given any Markov policy $\pi$ and its corresponding occupancy measure $q$, there exists a set of $M$ greedy policies and a probability distribution $\{a_m\}_{m=1,\cdots,M}$ such that the mixed policy, which selects a greedy policy $\pi^m$ at the start of an episode with probability $a_m$ and subsequently follows it, has the same occupancy measure $q$ as the original policy $\pi$.*

*Proof.* To simplify the notation, we will prove the lemma for the case where $|\mathcal{D}_{h,x}(q)| \in \{1,2\}$, i.e., any stochastic decision takes two possible actions and assume $\mathcal{A} = \{0,1\}$. The extension to the general case is trivial.

Under a Markov policy $\{\pi_h\}_{h=1}^H$, the actions are independently chosen given state $x$ and step $h$. Suppose we will execute the Markov policy for $K$ episodes. We will generate $K$ matrices $\{B_k\}_{k=1}^K$ of size $H \times S$ such that $B_k(h,x)$ is a realization of a random variable with distribution $\pi_h(\cdot|x)$. All these values are independently generated. Now to execute policy $\pi$ at episode $k$, at state $x$ and step $h$, the agent takes action $a$ such that $B_k(h,x) = a$. This is statistically the same as sampling an action using $\pi_h(\cdot|x)$ when reaching state $x$ at step $h$.

We note that each binary matrix $B_k$ corresponds to a greedy policy from the $M$ greedy policies and vice versa. Furthermore, the binary matrix associated with greedy policy $\pi^m$ is generated with probability

$$\alpha_m = \prod_{h,x} \left( \sum_{a \in \mathcal{D}_{h,x}(q)} \pi_h(a|x)\pi_h^m(a|x) \right),$$

because

$$\sum_{a \in \mathcal{D}_{h,x}(q)} \pi_h(a|x)\pi_h^m(a|x) = \sum_{a \in \mathcal{D}_{h,x}(q)} \pi_h(a|x)\mathbb{I}(\pi_h^m(a|x) = 1),$$

which is the probability that action selected by the greedy policy $\pi^m$ is also selected under policy $\pi$. Therefore, if we consider a mixed policy that chooses policy $\pi^m$ with probability $a_m$, then it is statistically the same as policy $\pi$ and has the same occupancy measure $q$. $\square$

## C PROOF OF THEOREM 1 (PRUNING)

In this part, we are going to show the detailed proof of Pruning.

**Theorem 1.** *Let $\mathcal{D}^* = \{(h,x,a) : \pi_h^*(a|x) > 0\}$ and $\tilde{\mathcal{D}} = \left\{ (h,x,a) : \frac{\tilde{N}_h(x,a)}{\sqrt{K}} \geq \frac{\epsilon}{2} \right\}$. Under Assumptions 1 to 3, after policy pruning, we have*

$$\Pr\left( \tilde{\mathcal{D}}_{h,x} = \mathcal{D}_{h,x}(q^*), \ \forall(h,x) \right) = 1 - \tilde{\mathcal{O}}\left( K^{-0.1} \right).$$

*Proof.* At the end of the first phase, i.e., $\sqrt{K}$ episodes, we consider a mixed policy $\hat{\pi}$ that selects the policy used in the $k$th episode, $\pi_k$, with probability $1/\sqrt{K}$. We assume that all constraints are satisfied under $\hat{\pi}$, which occurs with probability $1 - \mathcal{O}(K^{-2})$. The reward value function of the policy $\hat{\pi}$ is

$$V_1^{\hat{\pi}} = \frac{1}{\sqrt{K}} \sum_{k=1}^{\sqrt{K}} V_1^{\pi_k} \tag{15}$$

and $V^{\pi^*} - V_1^{\hat{\pi}} \geq 0$ because the constraints are satisfied under $\hat{\pi}$. Note that policy $\hat{\pi}$ is not a Markov policy. We next prove that the occupancy measure induced by policy $\hat{\pi}$ is a valid solution to the LP problem:

$$\sum_a q_h^{\hat{\pi}}(x,a) = \sum_a \frac{1}{\sqrt{K}} \sum_{k=1}^{\sqrt{K}} q_h^{\pi_k}(x,a)$$

$$= \frac{1}{\sqrt{K}} \sum_{k=1}^{\sqrt{K}} \sum_a q_h^{\pi_k}(x,a)$$

$$= \frac{1}{\sqrt{K}} \sum_{k=1}^{\sqrt{K}} \sum_{x',a'} q_{h-1}^{\pi_k}(x',a') \mathbb{P}_h(x|x',a')$$

$$= \sum_{x',a'} \mathbb{P}_h(x|x',a') \frac{1}{\sqrt{K}} \sum_{k=1}^{\sqrt{K}} q_{h-1}^{\pi_k}(x',a')$$

$$= \sum_{x',a'} \mathbb{P}_h(x|x',a') q_{h-1}^{\hat{\pi}}(x',a').$$

Besides, it is easy to verify that $\forall h, x, a, q_h^{\hat{\pi}}(x,a) \geq 0$. Thus the policy $\hat{\pi}$ is a valid policy for the LP problem.

Recall that $\mathcal{D}_{h,x}(q) = \{a : q_h(x,a) > 0\}$. We have $\tilde{\mathcal{D}}_{h,x}(q) = \mathcal{D}_{h,x}(q^*)$, $\forall(h,x)$ is equivalent to $\mathcal{D}^* = \tilde{\mathcal{D}}$.

We further define event

$$\mathcal{E} = \left\{ \exists(h,x,a) \in \mathcal{D}^*, \frac{\tilde{N}_h(x,a)}{\sqrt{K}} < \frac{\epsilon}{2} \right\},$$

i.e., the event that at the end of the pruning phase, the algorithm eliminates an action used by the optimal policy. Note that $\pi_k$'s are greedy policies ($\pi_{k,h}(a|x) \in \{0,1\}$). Therefore, we have

$$\tilde{N}_h(x,a) = \sum_{k=1}^{\sqrt{K}} \pi_{k,h}(a|x).$$

Thus, assuming this event $\mathcal{E}$ occurs, we can obtain

$$q_h^{\hat{\pi}}(x,a)$$

$$= \frac{1}{\sqrt{K}} \sum_{k=1}^{\sqrt{K}} q_h^{\pi_k}(x,a)$$

$$= \frac{1}{\sqrt{K}} \sum_{k=1}^{\sqrt{K}} \left( \sum_{x',a'} q_{h-1}^{\pi_k}(x',a') \mathbb{P}_h(x|x',a') \right) \pi_{k,h}(a|x)$$

$$\leq \frac{1}{\sqrt{K}} \sum_{k=1}^{\sqrt{K}} \pi_{k,h}(a|x)$$

$$= \frac{1}{\sqrt{K}} \tilde{N}_h(x,a)$$

$$< \frac{\epsilon}{2}.$$

According to Assumption 1, we have

$$q_h^{\pi^*}(x,a) \geq \epsilon \quad \forall(h,x,a) \in \mathcal{D}^*, \tag{16}$$

which implies $||q^{\hat{\pi}} - q^*||_1 \geq \frac{\epsilon}{2}$. According to Assumption 2, we have either

$$V_1^{\pi^*} - V_1^{\hat{\pi}} \geq c_v \frac{\epsilon}{2} \quad \text{(Case 1)} \tag{17}$$

or

$$W_1^{\pi^*,n} - W_1^{\hat{\pi},n} \geq c_w \frac{\epsilon}{2} \text{ for some } n \quad \text{(Case 2)}. \tag{18}$$

Therefore, we have

$$\Pr(\mathcal{E}) \leq \Pr\left( \sqrt{K} \left( V_1^{\pi^*} - V^{\hat{\pi}} \right) \geq c_v \frac{\epsilon}{2} \sqrt{K} \right)$$

$$+ \Pr\left( \exists n \in [N], \sqrt{K} \left( W_1^{\pi^*,n} - W_1^{\hat{\pi},n} \right) \geq c_w \frac{\epsilon}{2} \sqrt{K} \right).$$

Based on Lemma 3's result on regret and the Markov inequality, we have

$$\Pr\left(\sqrt{K}\left(V_1^{\pi^*} - V^{\hat{\pi}}\right) \geq c_v \frac{\epsilon}{2}\sqrt{K}\right) \leq \frac{c_1 \sqrt{K}^{0.8}}{c_v \frac{1}{2}\epsilon\sqrt{K}} = \frac{2c_1}{c_v \epsilon K^{0.1}}.$$

From Lemma 3's result on constraint violation, the high probability bound, we have

$$\Pr\left(\exists n \in [N], \sqrt{K}\left(W_1^{\pi^*,n} - W_1^{\hat{\pi},n}\right) \geq c_w \frac{\epsilon}{2}\sqrt{K}\right)$$

$$\leq \Pr\left(\exists n \in [N], \sqrt{K}\left(\rho^{(n)} - W_1^{\hat{\pi},n}\right) \geq c_w \frac{\epsilon}{2}\sqrt{K}\right)$$

$$= \mathcal{O}\left(\frac{1}{K}\right).$$

Thus for sufficiently large $\sqrt{K}$, $\Pr(\mathcal{E}) = \mathcal{O}\left(K^{-0.1}\right)$, i.e., with probability $1 - \mathcal{O}\left(K^{-0.1}\right)$, we have

$$\mathcal{D}^* \subseteq \tilde{\mathcal{D}}.$$

Now define event $\mathcal{E}' = \left\{\exists(h,x,a) \notin \mathcal{D}^*, \frac{N_h(x,a)}{\sqrt{K}} \geq \frac{\epsilon}{2}\right\}$. Similar to equation 16 and based on Assumption 3, we can obtain

$$q_h^{\hat{\pi}}(x,a) = \frac{1}{\sqrt{K}}\sum_{k=1}^{\sqrt{K}} q_h^{\pi_k}(x,a) \geq p_{\min}\frac{1}{\sqrt{K}}\tilde{N}_h(x,a) \geq \frac{\epsilon p_{\min}}{2}. \tag{19}$$

Since $q_h^{\pi^*}(x,a) = 0$ for $(h,x,a) \notin \mathcal{D}^*$, $\|q^{\hat{\pi}} - q^{\pi^*}\|_1 \geq \frac{\epsilon p_{\min}}{2}$. Similar to the analysis on $\mathcal{E}$, we obtain

$$\Pr(\mathcal{E}') = \mathcal{O}\left(K^{-0.1}\right). \tag{20}$$

In other words, with probability $1 - \mathcal{O}\left(K^{-0.1}\right)$, we have

$$\tilde{\mathcal{D}} \subseteq \mathcal{D}^*,$$

which completes the proof. $\qquad\square$

## D  PROOF OF THEOREM 2 (REFINEMENT)

**Theorem 2.** *Assume $\tilde{\mathcal{D}} = \mathcal{D}^*$ after policy pruning. Under Assumptions 1 and 3, with probability $1 - \tilde{\mathcal{O}}\left(\frac{1}{\sqrt{K}}\right)$, the regret and constraint violation during the policy refinement phase are both $\mathcal{O}\left(H\sqrt{K}\right)$.*

*Proof.* Recall that in Lemma 2, we have shown that there exists a mixed policy of $M$ greedy policies defined by $\mathcal{D}_{h,x}^*$ that has the same occupancy measure as that under the optimal policy, and $\{\alpha_m^*\}$ are the associated weights.

Recall that the policy refinement consists of $\sqrt{K}$ rounds. Let $\{\alpha_{t,m}\}_{m=1,\cdots,M}$ be the weights used in round $t$. Then in the $t$th round, greedy policy $\pi^m$ is used for $\alpha_{t,m}\sqrt{K}$ episodes, where $\alpha_{t,m}$ is the optimal solution to Decomposition-Opt equation 11.

First, we will bound the estimation errors of the reward and utility value functions. Recall that PRI uses $\epsilon'\sqrt{K}$ episodes in each round to estimate the reward value function and the utility value functions instead of all episodes because $\{\alpha_{t,m}\}$ are random variables correlated with the estimated value functions from the previous round. At the beginning of round $t$, we have $(t-1)\epsilon'\sqrt{K}$ samples from the previous round. Indexing the samples by $k'$, we have

$$\bar{W}_1^{\pi^m,n} = \frac{\sum_{k'=1}^{(t-1)\epsilon'\sqrt{K}} W_{k',1}^{\pi^m,n}}{(t-1)\epsilon'\sqrt{K}}. \tag{21}$$

Define

$$\delta W_1^{\pi^m, n} = \bar{W}_1^{\pi^m, n} - W_1^{\pi^m, n} \tag{22}$$

$$= \frac{\sum_{k'=1}^{(t-1)\epsilon'\sqrt{K}} \left( W_{k',1}^{\pi^m, n} - W_1^{\pi^m, n} \right)}{(t-1)\epsilon'\sqrt{K}}. \tag{23}$$

Since $W_{k',1}^{\pi^m, n} \in [0, H]$ are i.i.d. random variables, by the Azuma-Hoeffding inequality, we have

$$\Pr\left( \left| \delta W_1^{\pi^m, n} \right| \leq \sqrt{\frac{2H^2 \log\left( (t-1)\epsilon'\sqrt{K} \right)}{\epsilon'(t-1)\sqrt{K}}} \right) \tag{24}$$

$$\geq 1 - \frac{2}{(t-1)\epsilon'\sqrt{K}}. \tag{25}$$

Similarly, defining

$$\delta V_1^{\pi^m} = \bar{V}_1^{\pi^m} - V_1^{\pi^m} = \frac{\sum_{k'=1}^{(t-1)\epsilon' K^\alpha} \left( V_{k',1}^{\pi^m} - V_1^{\pi^m} \right)}{(t-1)\epsilon'\sqrt{K}},$$

we have

$$\Pr\left( \left| \delta V_1^{\pi^m} \right| \leq \sqrt{\frac{2H^2 \log\left( (t-1)\epsilon'\sqrt{K} \right)}{\epsilon'(t-1)\sqrt{K}}} \right) \tag{26}$$

$$\geq 1 - \frac{2}{(t-1)\epsilon'\sqrt{K}}. \tag{27}$$

Therefore, with probability at least $1 - \frac{2M}{(t-1)\epsilon'\sqrt{K}}$,

$$\left| \sum_{m=1}^M \alpha_m^* \bar{W}_1^{\pi_m, n} - \rho^{(n)} \right| = \left| \sum_{m=1}^M \alpha_m^* \left( W_1^{\pi_m, n} + \delta W_1^{\pi_m, n} \right) - \rho^{(n)} \right|$$

$$\leq \sum_{m=1}^M \alpha_m^* \left| \delta W_1^{\pi_m, n} \right|$$

$$\leq \sqrt{\frac{2H^2 \log\left( (t-1)\epsilon'\sqrt{K} \right)}{\epsilon'(t-1)\sqrt{K}}}.$$

In other words, $\{a_m^*\}$ is a feasible solution to Decomposition-Opt equation 11 with a high probability, which implies that Decomposition-Opt equation 11 has a solution with a high probability.

We now consider $\{a_{t,m}\}$ and the regret and constraint violation in round $t$. If $\{a_m^*\}$ is a feasible solution to Decomposition-Opt equation 11, then

$$\sum_{m=1}^M \alpha_{t,m} \sqrt{K} V_1^{\pi^m}$$

$$= \sum_{m=1}^M \alpha_{t,m} \sqrt{K} \left( \bar{V}_1^{\pi^m} - \delta V_1^{\pi^m} \right)$$

$$\geq \sqrt{K} \left( \sum_{m=1}^M \alpha_{t,m} \bar{V}_1^{\pi^m} \right) - \sqrt{K} \max_m \left| \delta V_1^{\pi^m} \right|$$

$$\geq_{(a)} \sqrt{K} \left( \sum_{m=1}^{M} \alpha_m^* \left( V_1^{\pi^m} + \delta V_1^{\pi^m} \right) \right) - \sqrt{K} \max_m |\delta V_1^{\pi^m}|$$

$$\geq \sqrt{K} \left( \sum_{m=1}^{M} \alpha_m^* V_1^{\pi^m} \right) - 2\sqrt{K} \max_m |\delta V_1^{\pi^m}|$$

$$= \sqrt{K} V_1^{\pi^*} - 2\sqrt{K} \max_m |\delta V_1^{\pi^m}|$$

$$\geq \sqrt{K} \left( V_1^{\pi^*} - 2\sqrt{\frac{2H^2 \log \left( (t-1)\epsilon' \sqrt{K} \right)}{\epsilon'(t-1)\sqrt{K}}} \right),$$

where $(a)$ holds because $\{a_{t,m}\}_m$ is the optimal solution to Decomposition-Opt equation 11. In other words, with a high probability, the regret is bounded by

$$2\sqrt{\frac{2\sqrt{K}H^2 \log \left( (t-1)\sqrt{K} \right)}{\epsilon'(t-1)}}. \tag{28}$$

Thus, with probability

$$\prod_{t=2}^{\sqrt{K}} \left( 1 - \frac{2}{(t-1)\epsilon'\sqrt{K}} \right) \geq 1 - \frac{2\log K}{\epsilon'\sqrt{K}} \tag{29}$$

regret in round $t$ is bounded by equation 28 for all $t$. Therefore, the regret during policy refinement is bounded by

$$2H\sqrt{K} + \sum_{t=3}^{\sqrt{K}} 2\sqrt{\frac{2\sqrt{K}H^2 \log \left( (t-1)\sqrt{K} \right)}{\epsilon'(t-1)}}$$

$$\leq 2H\sqrt{K} + 2\sqrt{\frac{2KH^2 \log K}{\epsilon'}} \sum_{t=3}^{\sqrt{K}} \sqrt{\frac{1}{(t-1)\sqrt{K}}}$$

$$\leq 2H\sqrt{K} + 2\sqrt{\frac{2\sqrt{K}H^2 \log K}{\epsilon'}} \int_{t=1}^{\sqrt{K}} \sqrt{\frac{1}{t}} dt$$

$$\leq 2H\sqrt{K} + 2\sqrt{\frac{2KH^2 \log K}{\epsilon'}}$$

$$= \mathcal{O}(H\sqrt{K}).$$

The analysis is the same for the constraint violation. $\qquad \square$

## E  PROOF OF THEOREM 3 (IDENTIFICATION)

**Theorem 3.** *Assume $\tilde{\mathcal{D}} = \mathcal{D}^*$ after policy pruning. Under Assumptions 1 and 3, with probability $1 - \tilde{\mathcal{O}}\left( \frac{1}{K} \right)$, the regret and constraint violation during the policy identification phase are both $\mathcal{O}\left( \sqrt{K} \right)$. Furthermore, $|\tilde{\pi}_h(a|x) - \pi_h^*(a|x)| = \mathcal{O}\left( \frac{1}{\sqrt{K}} \right)$ if $0 < \pi_h^*(a|x) < 1$ and $\tilde{\pi}_h(a|x) = \pi_h^*(a|x)$ if $\pi_h^*(a|x) \in \{0,1\}$.*

*Proof.* Consider the $\{a_m\}$ obtained at the end of the refinement phase, and the mixed policy $\hat{\pi}$ defined by $\{a_m\}$. According to the proof of Theorem 2, we have with probability $1 - \mathcal{O}(K^{-1})$,

$$V_1^{\hat{\pi}} = \sum_{m=1}^{M} \alpha_m V_1^{\pi^m}$$

$$\geq \left( V_1^{\pi^*} - 2\sqrt{\frac{H^2 \log\left((t-1)\epsilon'K\right)}{\epsilon'K}} \right) \tag{30}$$

$$W_1^{\hat{\pi},n} = \sum_{m=1}^{M} \alpha_m W_1^{\pi^m,n}$$

$$\geq \left( W_1^{\pi^*,n} - 2\sqrt{\frac{H^2 \log\left((t-1)\epsilon'K\right)}{\epsilon'K}} \right) \quad \forall n. \tag{31}$$

Therefore, the regret and constraint violation during the identification phase, which includes $K$ episodes, are both $\mathcal{O}(H\sqrt{K})$.

When both equation 30 and equation 31 hold, under Assumption 2, we have

$$\|q^{\hat{\pi}} - q^{\pi^*}\|_1 = \mathcal{O}(1/\sqrt{K}).$$

For any $(h, x, a)$ such that $0 < \pi_h^*(x|a) < 1$,

$$\mathbb{E}\left[ \sum_{k=1}^{\lceil a_m K \rceil} \mathbb{I}(x_{k,h} = x, a_{k,h} = a) \right] = \alpha_m q_h^{\pi^m}(x, a)K,$$

which implies that

$$\Pr\left( \left| \sum_{k=1}^{\lceil a_m K \rceil} \mathbb{I}_{(x_{k,h}=x, a_{k,h}=a)} - \alpha_m q_h^{\pi^m}(x, a)K \right| \leq \sqrt{K \log K} \right)$$

$$= 1 - \mathcal{O}\left( \frac{1}{K} \right)$$

according to the Azuma-Hoeffding inequality. Define event

$$\Phi = \left\{ \left| \frac{\sum_{k=1}^{K} \mathbb{I}_{(x_{k,h}=x, a_{k,h}=a)}}{K} - \sum_m \alpha_m q_h^{\pi^m}(x, a) \right| \leq M\sqrt{\frac{\log K}{K}} \right\}. \tag{32}$$

We have

$$\Pr\left( \Phi \right) = 1 - \mathcal{O}\left( \frac{1}{K} \right).$$

Define $\tilde{q}_h(x, a) = \frac{N_h(x,a)}{K}$, which is the empirical occupant measure under policy $\hat{\pi}$. Note that

$$N_h(x, a) = \sum_{k=1}^{K} \mathbb{I}(x_{k,h} = x, a_{k,h} = a)$$

and

$$q_h^{\hat{\pi}}(x, a) = \sum_m \alpha_m q_h^{\pi^m}(x, a).$$

Therefore, we have with probability $1 - \mathcal{O}(1/K)$,

$$\|\tilde{q} - q^{\pi^*}\|_1 = \mathcal{O}(1/\sqrt{K}),$$

which implies that

$$\|\tilde{\pi} - \pi^*\|_1 = \mathcal{O}(1/\sqrt{K}).$$

Furthermore, since $\tilde{\mathcal{D}} = \mathcal{D}^*$, we immediately have $\tilde{\pi}_h(a|x) = \pi_h^*(a|x)|$ if $\pi_h^*(a|x) \in \{0, 1\}$. $\qquad\square$

## F   EXTENSION TO CMDPS WITH MULTIPLE OPTIMAL SOLUTIONS

When the optimal solution to the CMDP is not unique, or the RL agent does not know whether the CMDP has a unique solution or not, the agent adds Multi-Solution Pruning after the pruning phase in PRI to keep one and only one optimal policy belonging to $\Pi^{*,e}$. Recall that after the pruning

phase, the action space for state $x$ and step $h$, denoted by $\mathcal{A}_{h,x}$, is limited to $\mathcal{A}_{h,x} = \tilde{D}_{h,x}$. The key idea of the multi-solution pruning algorithm is to evaluate each stochastic decision $(h', x')$ such that $|\mathcal{A}_{h',x'}| > 1$. The algorithm first decides whether some of the actions in $\mathcal{A}_{h',x'}$ can be removed, e.g., $a'$, while retaining at least one optimal policy with the following action space:

$$\otimes_{(h,x) \neq (h',x')} \mathcal{A}_{h,x} \otimes (\mathcal{A}_{h',x'} \setminus \{a'\}).$$

This is done by running Triple-Q with the above action space for $K^{0.25}$ episodes. If the regret is small, then with a high probability, at least one of the optimal policies is retained so we can remove action $a'$ from $\mathcal{A}_{h',x'}$. The detailed algorithm is presented in Algorithm 2.

If the regret is large, then any optimal policy in $\otimes_{(h,x)} \mathcal{A}_{h,x}$ has to use action $a'$ in state $x'$ at step $h'$. Mulit-Solution Pruning next determines whether using $a'$ alone is sufficient, i.e., whether an optimal policy is retained in the following action space

$$\otimes_{(h,x) \neq (h',x')} \mathcal{A}_{h,x} \otimes (\mathcal{A}_{h',x'} = \{a'\}).$$

This is again done by running Triple-Q with the above action space for $K^{0.25}$ episodes. If the regret is small, then with a high probability, one optimal policy takes a greedy decision at $(h', x')$ with action $a'$; otherwise, the algorithm keeps $a'$ in $\mathcal{A}_{h',x'}$ and moves to a different action in $\mathcal{A}_{h',x'}$. Note that we use Triple-Q for $K^{0.25}$ episodes each time, instead of $\sqrt{K}$ episodes, because it is easier to learn whether an optimal policy exists than learning the actual optimal policy.

---

**Algorithm 2** Multi-Solution Pruning

---

Set $v^*$ to be the average cumulative reward received over the $\sqrt{K}$ episodes under the policy pruning phase of PRI.

Set flag$(h, x) \leftarrow 0$ for all $(h, x)$ such that $|\tilde{\mathcal{D}}_{h,x}| > 1$.

**while** $\exists (h, x)$ such that $|\tilde{\mathcal{D}}_{h,x}| > 1$ and flag$(h, x) = 0$ **do**

  Select $(h', x')$ such that $|\tilde{\mathcal{D}}_{h',x'}| > 1$ and flag$(h', x') = 0$ with the smallest $h'$. Ties are broken arbitrarily.

  flag$(h', x') \leftarrow 1$

  **for** $a' \in \tilde{\mathcal{D}}_{h',x'}$ **do**

    Reset Triple-Q and run it for $K^{0.25}$ episodes with $\tilde{D}_{h,x}$ $((h, x) \neq (h', x'))$ and $\tilde{D}_{h',x'} \setminus \{a'\}$ as the action spaces while counting $\tilde{N}_h(x, a)$ as in policy pruning. Record the average cumulative reward $\tilde{v}$ and average cumulative utilities $\tilde{w}^n$.

    **if** $v^* - \tilde{v} \leq \frac{2}{K^{0.03}}$ and $\tilde{w}^n \geq \rho^{(n)}$ for all $n$ **then**

      Update $\tilde{\mathcal{D}}_{h,x} = \left\{ a : \frac{\tilde{N}_h(x,a)}{K^{0.25}} \geq \frac{\epsilon}{2} \right\}$ for all $(h, x)$.

    **else**

      Run Triple-Q for $K^{0.25}$ episodes with action space $\tilde{D}_{h,x}$ for $(h, x) \neq (h', x')$ and $\{a'\}$ for $(h', x')$. Record the average cumulative reward $\tilde{v}$

      **if** $v^* - \tilde{v} \leq \frac{2}{K^{0.03}} + 2\sqrt{\frac{2H^2 \log K^{0.25}}{K^{0.25}}}$ and $\tilde{w}^n \geq \rho^{(n)}$ for all $n$ **then**

        Update $\tilde{\mathcal{D}}_{h,x} = \left\{ a : \frac{\tilde{N}_h(x,a)}{K^{0.25}} \geq \frac{\epsilon}{2} \right\}$ for all $(h, x)$.

---

### F.1 PROOF OF THEOREM 4 (MULTIPLE-SOLUTION PRUNING)

**Theorem 4.** *Under Assumption 4 and 5, with probability $1 - \mathcal{O}(1/K^{0.02})$, for sufficiently large $K$, multi-solution pruning outputs a unique optimal policy with at most $N$ stochastic decisions. The regret and constraint violation during multi-solution pruning are bounded by $H^2 SAK^{0.25}$ with probability one.*

*Proof.* We first consider the $K^{0.25}$ episodes after $a'$ is removed from $\mathcal{A}_{h',x'}$. Consider the case that there still exists an optimal policy after removing the action. In this case, we will show that $v^* - \tilde{v} \leq \frac{2}{K^{0.03}}$ with a high probability. Define

$$\bar{V}_1 = \frac{1}{K^{0.25}} \sum_{k=1}^{K^{0.25}} V_1^{\pi_k},$$

where $\pi_k$ is the policy used in the $k$th episode.

Note that

$$v^* - \tilde{v} = v^* - V_1^{\pi^*} + V_1^{\pi^*} - \bar{V}_1 + \bar{V}_1 - \tilde{v}.$$

We next bound the three terms $v^* - V_1^{\pi^*}$, $V_1^{\pi^*} - \bar{V}_1$, and $\bar{V}_1 - \tilde{v}$ individually.

Let $v_{k,1}$ be the cumulative reward received in episode $k$ and $V_1^{\pi_k}$ be the reward value function. Note that

$$X_\tau = \sum_{k=1}^\tau \left( v_{k,1} - V_1^{\pi_k} \right)$$

is a Martingale. By Azuma's inequality, we have

$$\Pr\left( \left| \tilde{v} - \bar{V}_1 \right| \le \sqrt{\frac{2H^2 \log K^{0.25}}{K^{0.25}}} \right) \ge 1 - \frac{1}{2K^{0.25}}. \tag{33}$$

A similar argument yields that

$$\Pr\left( \left| v^* - \bar{V}_1^{\pi^*} \right| \le \sqrt{\frac{2H^2 \log K^{0.5}}{K^{0.5}}} \right) \ge 1 - \frac{1}{2K^{0.5}}. \tag{34}$$

We next bound $V_1^{\pi^*} - \bar{V}_1$ based on Lemma 3 and the Markov inequality. First, based on the Markov inequality, we have

$$\Pr\left( V_1^{\pi^*} - \bar{V}_1 \ge K^{-0.03} \,\middle|\, V_1^{\pi^*} \ge \bar{V}_1 \right) \le \frac{\mathbb{E}\left[ V_1^{\pi^*} - \bar{V}_1 \,\middle|\, V_1^{\pi^*} \ge \bar{V}_1 \right]}{K^{-0.03}}.$$

Note that we have

$$\mathbb{E}\left[ V_1^{\pi^*} - \bar{V}_1 \,\middle|\, V_1^{\pi^*} \ge \bar{V}_1 \right] = \frac{\mathbb{E}\left[ V_1^{\pi^*} - \bar{V}_1 \right] - \Pr\left( V_1^{\pi^*} < \bar{V}_1 \right) \mathbb{E}\left[ V_1^{\pi^*} - \bar{V}_1 \,\middle|\, V_1^{\pi^*} < \bar{V}_1 \right]}{\Pr\left( V_1^{\pi^*} \ge \bar{V}_1 \right)} \tag{35}$$

$$\le \frac{\frac{c_1 K^{0.2}}{K^{0.25}} + H \Pr\left( V_1^{\pi^*} < \bar{V}_1 \right)}{1 - \Pr\left( V_1^{\pi^*} < \bar{V}_1 \right)} \tag{36}$$

and

$$\Pr\left( V_1^{\pi^*} - \bar{V}_1 \ge K^{-0.03} \right) \tag{37}$$

$$= \Pr\left( V_1^{\pi^*} - \bar{V}_1 \ge K^{-0.03} \,\middle|\, V_1^{\pi^*} \ge \bar{V}_1 \right) \Pr\left( V_1^{\pi^*} \ge \bar{V}_1 \right) \tag{38}$$

$$+ \Pr\left( V_1^{\pi^*} - \bar{V}_1 \ge K^{-0.03} \,\middle|\, V_1^{\pi^*} < \bar{V}_1 \right) \Pr\left( V_1^{\pi^*} < \bar{V}_1 \right) \tag{39}$$

$$\le \Pr\left( V_1^{\pi^*} - \bar{V}_1 \ge K^{-0.03} \,\middle|\, V_1^{\pi^*} \ge \bar{V}_1 \right) \left( 1 - \Pr\left( V_1^{\pi^*} < \bar{V}_1 \right) \right) + \Pr\left( V_1^{\pi^*} < \bar{V}_1 \right) \tag{40}$$

$$= K^{0.03} \left( \frac{c_1 K^{0.2}}{K^{0.25}} + H \Pr\left( V_1^{\pi^*} < \bar{V}_1 \right) \right) + \Pr\left( V_1^{\pi^*} < \bar{V}_1 \right). \tag{41}$$

Note that when the constraints are satisfied, we have $V_1^{\pi^*} \ge \bar{V}_1$. Therefore, according Lemma 3, $\Pr\left( V_1^{\pi^*} < \bar{V}_1 \right) = \mathcal{O}(K^{-0.5})$, which implies that

$$\Pr\left( V_1^{\pi^*} - \bar{V}_1 \ge K^{-0.03} \right) = \mathcal{O}\left( K^{-0.02} \right). \tag{42}$$

Combining the inequality above with inequalities equation 33 and equation 34, we can conclude that

$$\Pr\left( v^* - \tilde{v} \le 2K^{-0.03} \right) = 1 - \mathcal{O}\left( K^{-0.02} \right). \tag{43}$$

Based on Lemma 3's result on constraint violation, we obtain

$$\Pr\left( v^* - \tilde{v} \le 2K^{-0.03}, \tilde{w}^n \ge \rho^{(n)} \,\forall n \right) = 1 - \mathcal{O}\left( K^{-0.02} \right), \tag{44}$$

On the other hand, if no optimal policy exists after removing $a'$, then we have $\forall \pi \in \Pi^{*,e}$, $q_{h'}^{\pi}(x', a') \geq \epsilon$. Let $\pi''$ be an optimal policy with action spaces

$$\otimes_{(h,x) \neq (h',x')} \mathcal{A}_{h,x} \otimes (\mathcal{A}_{h',x'} \setminus \{a'\}),$$

and suppose all constraints are satisfied under $\pi''$. Note that $\pi''$ is *not* an optimal policy for the original problem. We first have

$$v^* - \tilde{v} = v^* - V_1^{\pi^*} + V_1^{\pi^*} - V_1^{\pi''} + V_1^{\pi''} - \bar{V}_1 + \bar{V}_1 - \tilde{v}$$

Based on Assumptions 4 and 5, we have

$$\begin{aligned} V_1^{\pi^*} - V_1^{\pi''} &\geq c_v ||q^{\pi^*} - q^{\pi''}||_1 \\ &\geq c_v |q_{h'}^{\pi^*}(x', a') - q_{h'}^{\pi''}(x', a')| \\ &\geq c_v \epsilon. \end{aligned}$$

This holds because all constraints are satisfied under $\pi''$ under our assumption. Similar to the first case, we have

$$\Pr\left(\left|\tilde{v} - \bar{V}_1\right| \leq \sqrt{\frac{2H^2 \log K^{0.25}}{K^{0.25}}}\right) \geq 1 - \frac{1}{2K^{0.25}}. \tag{45}$$

and

$$\Pr\left(\left|v^* - \bar{V}_1^{\pi^*}\right| \leq \sqrt{\frac{2H^2 \log K^{0.5}}{K^{0.5}}}\right) \geq 1 - \frac{1}{2K^{0.5}}. \tag{46}$$

Furthermore, according to Lemma 3,

$$\Pr\left(V_1^{\pi''} - \bar{V}_1 \geq 0\right) \geq 1 - \mathcal{O}\left(\frac{1}{K^{0.5}}\right) \tag{47}$$

because the constraint is satisfied with probability $1 - \mathcal{O}\left(\frac{1}{K^{0.5}}\right)$.

Summarizing the results above, using the union bound, we conclude that with probability $1 - \mathcal{O}\left(K^{-0.02}\right)$, we have

$$v^* - \tilde{v} \geq c_v \epsilon - 2\sqrt{\frac{2H^2 \log K^{0.25}}{K^{0.25}}} > 2K^{-0.03}$$

for sufficiently large $K$ if an optimal policy is not retained. If none of the policies formed by action spaces

$$\otimes_{(h,x) \neq (h',x')} \mathcal{A}_{h,x} \otimes (\mathcal{A}_{h',x'} \setminus \{a'\})$$

can satisfy the constraints, then it can be easily shown that $\tilde{w}^n < \rho^{(n)}$ with probability $1 - \mathcal{O}(K^{-0.25})$ for some $n$.

If $a'$ is deemed to be necessary, Mulit-Solution Pruning next determines whether using $a'$ alone is sufficient, i.e., can stochastic decision $(h', x')$ become greedy without losing optimality? The algorithm runs Triple-Q with action space $\{a'\}$ for $(h', x')$. If there exists an optimal policy $\pi$ with $\pi_{h'}(a'|x') = 1$, then following the same analysis above, we have with probability at least $1 - \mathcal{O}\left(K^{-0.02}\right)$,

$$v^* - \tilde{v} \leq \frac{2}{K^{0.03}}.$$

Otherwise, according to the Assumption 4, $\forall \pi^* \in \Pi^{*,e}$, there exists another action $a'' \neq a'$ such that $q_{h'}^{\pi^*}(x', a'') \geq \epsilon$. Because any optimal policy can be represented as a linear combination of those policies in $\Pi^{*,e}$, we have that for any optimal policy $\pi^*$, $\sum_{a \neq a'} q_{h'}^{\pi^*}(x', a) \geq \epsilon$. Letting $\pi''$ be an optimal policy with action spaces

$$\otimes_{(h,x) \neq (h',x')} \mathcal{A}_{h,x} \otimes (\mathcal{A}_{h',x'} = \{a'\}),$$

which satisfies all constraints, we have

$$\sum_{a \neq a'} q_{h'}^{\pi^*}(x', a) - q_{h'}^{\pi''}(x', a) \geq \epsilon.$$

Thus, according to Assumption 5,

$$V_1^{\pi*} - V_1^{\pi''} \geq c_v ||q^{\pi*} - q^{\pi''}||_1 \geq c_v \epsilon$$

because $\pi''$ satisfies all constraints.

The remaining analysis is identical to case when $a'$ is removed from the action space $\mathcal{A}_{h',x'}$. If none of the policies formed by action spaces

$$\otimes_{(h,x) \neq (h',x')} \mathcal{A}_{h,x} \otimes (\mathcal{A}_{h',x'} = \{a'\})$$

can satisfy the constraints, then it can be easily shown that $\tilde{w}^n < \rho^{(n)}$ with probability $1 - \mathcal{O}(K^{-0.25})$ for some $n$.

After the algorithm goes through all action space $\mathcal{A}_{h,x}$, with probability $1 - \mathcal{O}(1/K^{0.02})$, we obtain action space

$$\otimes_{(h,x)} \mathcal{A}_{h,x}$$

such that none of the stochastic decision can be reduced to a greedy decision without losing optimality. Since any optimal policy can be written as a linear combination of optimal policies associated with extreme points, and any combination of two optimal policy only increases the number of stochastic decisions. Therefore, we conclude that the optimal policy induced by

$$\otimes_{(h,x)} \mathcal{A}_{h,x}$$

is an extreme point and is unique. Besides, it is easy to verify that the regret and constraint violation of multiple solution pruning are bounded by $H^2 SAK^{0.25}$ because it takes at most $HSAK^{0.25}$ episodes to finish the algorithm. $\qquad \square$

## G EXTENSION TO CONSTRAINT VIOLATION WITHOUT EPISODE-WISE CANCELLATION

While the constraint violation defined in (5) allows the cancellation across episodes, we can guarantee $O(\sqrt{K} \log K)$ violation when the cancellation is not allowed, as defined in (5) in Efroni et al. (2020), by making some minor modification to the algorithm. In particular, in the policy refinement and identification phases, instead of using the $M$ greedy policy in a round-robin fashion, we can use a mixed policy that chooses policy $m$ with probability $\alpha_m$. With that modification, we will have

- The Pruning phase consists at most $\sqrt{K}$ episodes, so resulting in at most $O(\sqrt{K})$ violation.

- During the refinement phase, based on inequality (27), we have with a high probability that the mixed policy used in the $t$th iteration has a regret bounded by $O\left(\frac{\log K}{(t-1)\sqrt{K}}\right)$ for $t \geq 2$. The same result holds for constraint violation, i.e., with a high probability, the constraint violation of the policy used in the $t$th iteration is bounded by $O\left(\frac{\log K}{(t-1)\sqrt{K}}\right)$. The $t$th iteration includes $\sqrt{K}$ episodes and the refinement phase includes $\sqrt{K}$ iterations. Therefore, with a high probability, the total violation without canceling across episodes is $O(\sqrt{K} \log K)$.

- The mixed policy used in the policy identification phase has a constraint violation bound of $O\left(\frac{\log K}{\sqrt{K}}\right)$. The policy is used for $K$ episodes, so the violation is bounded by $O(\sqrt{K} \log K)$.

Therefore, the total violation is bounded by $\tilde{O}(\sqrt{K})$ by using definition (5) in Efroni et al. (2020).

# H  SIMULATIONS

## H.1  SYNTHETIC CMDP

In the systehtic CMDP, we choose $|\mathcal{S}| = 3, |\mathcal{A}| = 3, H = 3$. The detailed parameters of the CMDP in the first experiment are shown in Table 2, 3 and 4.

We executed the pruning phase (Triple-Q) over $100,000$ episodes, followed by the refinement phase over $1,000,000$ episodes. Since the problem has only one constraint, the optimal policy has only one stochastic decision, which can be decided by evaluating the frequencies of two greedy policies. Thus, phase 3 is not necessary for this specific environment.

Table 2: Transition Kernels (the rows represent (previous state, action) and the columns represent (step, next state)).

|       | (1,1)      | (1,2)      | (1,3)      | (2,1)      | (2,2)      | (2,3)      | (3,1)      | (3,2)      | (3,3)      |
|-------|------------|------------|------------|------------|------------|------------|------------|------------|------------|
| (1,1) | 0.3112981  | 0.35107633 | 0.27041442 | 0.42626645 | 0.04822746 | 0.14663183 | 0.4031534  | 0.19783729 | 0.39831431 |
| (1,2) | 0.23314339 | 0.32491141 | 0.48360071 | 0.24246185 | 0.19021328 | 0.43972054 | 0.26457139 | 0.21435897 | 0.26256243 |
| (1,3) | 0.45555851 | 0.32401226 | 0.24598487 | 0.3312717  | 0.76155926 | 0.41364763 | 0.33227521 | 0.58780374 | 0.33912326 |
| (2,1) | 0.32676574 | 0.35320112 | 0.1300059  | 0.35453348 | 0.32114495 | 0.40817113 | 0.1762648  | 0.30097191 | 0.48437535 |
| (2,2) | 0.11092341 | 0.28034838 | 0.45655888 | 0.23441632 | 0.2847394  | 0.235718   | 0.17239783 | 0.37273618 | 0.08000908 |
| (2,3) | 0.56231085 | 0.3664505  | 0.41343525 | 0.4110502  | 0.39411565 | 0.35611087 | 0.65133738 | 0.32629191 | 0.43561556 |

Table 3: Rewards (the rows represent (state, action) and the columns represent step.)

|   | (1,1)      | (1,2)      | (1,3)      | (2,1)      | (2,2)      | (2,3)      | (3,1)      | (3,2)      | (3,3)      |
|---|------------|------------|------------|------------|------------|------------|------------|------------|------------|
| 1 | 0.5507979  | 0.70814782 | 0.29090474 | 0.51082761 | 0.89294695 | 0.89629309 | 0.12558531 | 0.20724388 | 0.0514672  |
| 2 | 0.44080984 | 0.02987621 | 0.45683322 | 0.64914405 | 0.27848728 | 0.6762549  | 0.59086282 | 0.02398188 | 0.55885409 |
| 3 | 0.25925245 | 0.4151012  | 0.28352508 | 0.69313792 | 0.44045372 | 0.15686774 | 0.54464902 | 0.78031476 | 0.30636353 |

Table 4: Utilities (the rows represent (state, action) and the columns represent step. )

|   | (1,1)      | (1,2)      | (1,3)      | (2,1)      | (2,2)      | (2,3)      | (3,1)      | (3,2)      | (3,3)      |
|---|------------|------------|------------|------------|------------|------------|------------|------------|------------|
| 1 | 0.22195788 | 0.38797126 | 0.93638365 | 0.97599542 | 0.67238368 | 0.90283411 | 0.84575087 | 0.37799404 | 0.09221701 |
| 2 | 0.6534109  | 0.55784076 | 0.36156476 | 0.2250545  | 0.40651992 | 0.46894025 | 0.26923558 | 0.29179277 | 0.4576864  |
| 3 | 0.86053391 | 0.5862529  | 0.28348786 | 0.27797751 | 0.45462208 | 0.20541034 | 0.20137871 | 0.51403506 | 0.08722937 |

## H.2  GRID WORLD

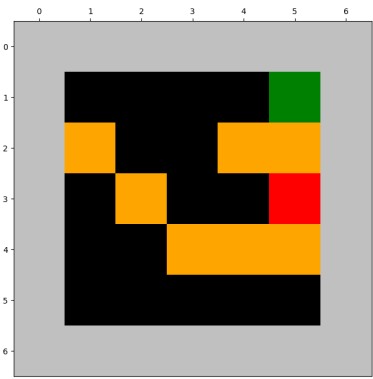

Figure 3: Grid World

As shown in Figure 3, the task of the agent is to go from the red grid point to the green grid point. The black grid points are the *safe* points over which the agent can move, and the yellow grid points

are obstacles. Moving over an obstacle incurs a penalty of one. The constraint is that the agent can incur only an average cost of $0.5$ or less. The agent can take six steps at maximum. The reward associated with reaching the destination is $1$, and the rewards for other locations, after six steps, are the Euclidean distance from the location to the destination (normalized by the longest distance). At each grid point, the agent has five actions to choose from: up, down, left, right, and stay, except at the boundary. The goal is to maximize the reward subject to the constraint.

During the experiment, we observed that policy pruning is much more efficient than the theoretical worst case. For this specific environment, the optimal policy should have $6 \times 5 \times 5 + 1 = 155$ nonzero $\pi_h^*(a|x)$'s. After the first phase (Triple-Q), we have roughly 200 (step, state, action) triples (here "roughly" considers the difference among different trials with different random seeds), associated with stochastic decisions to check and prune. Except for the two "necessary" decisions, which are stochastic decisions in the optimal policy, for all trials, the algorithm only checked two candidate triples and eliminated the rest candidate triples in the process.