# OpenReview forum: "Model-Free, Regret-Optimal Best Policy Identification in Online CMDPs"
_ICLR.cc/2024/Conference — Submitted to ICLR 2024_

### Official Review · Reviewer_y9UN · 2023-10-31

**Soundness:** 3 good
**Presentation:** 2 fair
**Contribution:** 3 good
**Rating:** 6
**Confidence:** 4

**Summary:**

This paper addresses the best policy identification (BPI) problem in online Constrained Markov Decision Processes (CMDPs), focusing on the development of a model-free algorithm with low regret that identifies an optimal policy with high probability. A new algorithm, Pruning-Refinement-Identification (PRI), is proposed, leveraging a newly discovered structural property of CMDPs named limited stochasticity. PRI ensures near-optimal policy output with a high probability and guarantees improved regret and constraint violation bounds in the tabular setting.

**Strengths:**

The result that there are at most $N$ states with a stochastic policy seems interesting. Here $N$ is the number of constraints.

**Weaknesses:**

1. Some assumptions of this paper seem a bit strong. In particular, this paper requires that each state-action pair can be visited with nontrivial probability $p_{min}$. Also, Assumptions 2 and 5 seem unnatural to me. When comparing with existing works, it would be nice to also compare with the assumptions made in these works.

2. Technically, this works seems a combination of Triple-Q with additional policy fine-tuning. It would be great to highlight the technical novelty. In particular, I wonder whether PRI can be used together with any online RL algorithm for CMDP with sublinear regret and constraint violation.

3. It would be great if the authors could conduct some simulation experiments that shows the the optimal policy is found, not just regret and constraint violation.

**Questions:**

1. When will Assumptions 2 and 5 hold?
2. Why do you need the policy to be unique in the first part of the theory?
3. In Theorem 4, do you get exactly an optimal policy or an approximate one as in Theorem 3.
4. Can you replace Triple-Q with any efficient CMDP algorithm as long as the regret and constraint violation are sublinear? Suppose the regret is $K^{\alpha}$ and constraint violation is $K^{\beta}$. What would be the error in learning the policy? I am asking because it seems that the proof of Theorem 3 only requires some regret and constraint violation results.

---

> ### Author Response · Authors · 2023-11-15
>
> Response:
>
> We thank the reviewer's positive comments. Our responses to your answers can be found below.
>
> **Weakness 1 and Q1.** Assumption 2 and 5 means when the policy deviates from the optimal policy, then either the reward value function or the utility value function, or both should also deviate from those under the optimal policy.  We believe this is a reasonable assumption and should hold for most CMDPs.
>
> **Weakness 2 and Q4.** As the reviewer correctly pointed out that the results of this paper hold when we replace Triple-Q with any online, model-free CMDP algorithm with sublinear regret and constraint violation.  PRI can be viewed as a ``meta-algorithm'' that builds on any model-free CMDP algorithm with sublinear regret and constraint violation to achieve  $\tilde{O}(\sqrt{K})$ regret and constraint violation and to output a single stochastic policy.
>
> **Weakness 3.** For the synthetic CMDP simulation result, we indeed show that the algorithm finds a policy with cumulative reward 1.57301 and cumulative utility 2.00008. The cumulative reward and cumulative utility under the optimal solution are 1.57306 and 2, respectively. This comparison is given at the beginning of page 9. We will further highlight it in the revision.
>
> **Q2.** The uniqueness is needed so that after the pruning phases, all stochastic decisions of the optimal policy are correctly identified. We note that this assumption is not needed and is relaxed in Section 6. We present this special case phase first for readability. Focusing on the special case
> where the problem has a unique optimal solution allows us to explain the key ideas behind the
> algorithm and makes the analysis easier to follow.
>
> **Q3.** The algorithm outputs a near-optimal solution. We will clarify this in the revision.

---

> > ### Comment · Reviewer_y9UN · 2023-11-22
> >
> > I would like to thank the authors for addressing my concerns. I am still not completely convinced by the explanation of the assumptions. Assumptions 2 and 5 not only mean that reward value function deviates from that of the optimal policy. It moreover suggests that the deviation should be lower bounded by the TV distance. This seems a very strong assumption and I cannot think of any MDP model that might satisfy this. Of course, bandit models satisfy these assumptions. It would be great if the authors could rigorously verify this assumption on some CMDP examples.

---

> ### Author Response · Authors · 2023-11-20
> **Any further questions/comments**
>
> We have uploaded a revision that incorporates your comments. The author-reviewer discussion ends in a few days, we would like to know whether you have any additional questions/comments we can address.

---

> ### Author Response · Authors · 2023-11-22
>
> Sorry for the confusion. We hope this reply can help you better understand the assumption we make.
> For any value function, according to the definition of occupancy measure, we have $$V^{\pi} = \sum_{h,x,a} r_h(x,a) q_h(x,a) = \mathbf{r} \cdot \mathbf{q^{\pi}}$$
> The assumption requires there should exist such $c_v$ that
> $$V^* - V^{\pi} = \mathbf{r} \cdot (\mathbf{q^*} - \mathbf{q^{\pi}}) \geq c_v||\mathbf{q^*} - \mathbf{q^{\pi}}||$$ $V$ is a linear function of $q$ with a deviation vector $\mathbf{r}$. That means the deviation for every specific direction must be a constant. Besides, with the limitation of constraints and legal transitions, the direction is restricted, which offers a minimum deviation. We assume the lower bound as $c_v$.
> For example, considering a MDP problem with $H=2, S=2, A=2$. Without loss of generality we only consider MDP while CMDP is the same. Suppose the reward function is: $r_1(1,1) = 1, r_1(1,2) = 2, r_1(2,1) = 1, r_1(2,2) = 2, r_2(1,1) = 1, r_2(1,2) = 2, r_2(2,1) = 1, r_2(2,2) = 2$. No matter what the transition kernel is, the optimal policy should always choose action 2 at any state at any step. Consider an occupancy measure $q_h^{\pi}(x,a)$ of any policy $\pi$,
> $$V^* - V^{\pi} = [\sum_{h,x} q_h^*(x,1) - q_h^{\pi}(x,1)] + [\sum_{h,x} 2(q_h^*(x,2) - q_h^{\pi}(x,2))] = -0.5 ||\mathbf{q^*} - \mathbf{q^{\pi}}|| + ||\mathbf{q^*} - \mathbf{q^{\pi}}|| = 0.5 ||\mathbf{q^*} - \mathbf{q^{\pi}}||$$
> For
> $$\sum_{h,x} q_h^*(x,1) - q_h^{\pi}(x,1) + \sum_{h,x} q_h^*(x,2) - q_h^{\pi}(x,2) = 0$$
> and
> $$|\sum_{h,x} q_h^*(x,1) - q_h^{\pi}(x,1)| + |\sum_{h,x} q_h^*(x,2) - q_h^{\pi}(x,2)| = ||\mathbf{q^*} - \mathbf{q^{\pi}}||.$$
> As long as $c_v \leq 0.5$, this MDP problem satisfies the assumption.
> On the other hand, actually we can relax the TV distance as $||q^* - q||_{\infty}$ and the proof will still hold. We hope this response can address your concern!

---

### Official Review · Reviewer_S5ah · 2023-11-01

**Soundness:** 3 good
**Presentation:** 3 good
**Contribution:** 3 good
**Rating:** 3
**Confidence:** 4

**Summary:**

The authors present a model-free algorithm for episodic constrained MDP that identifies the best policy within $\tilde{\mathcal{O}}(1/\sqrt{K})$ error and generates the optimal $\tilde{\mathcal{O}}(\sqrt{K})$ regret which vastly improves upon the best known model-free result.

**Strengths:**

The authors present a novel technique for proving the regret guarantees of CMDP. This might be useful in other constrained optimization setups. Overall, the paper reads well.

**Weaknesses:**

Please see the questions below.

**Questions:**

1. Episodic CMDP can be considered a special case of infinite-horizon average reward CMDP. An average reward CMDP can be transformed into an episodic CMDP by substituting $T=HK$ and augmenting a time index modulo $H$ in the state description. Therefore, Table 1 should also include the equivalent results obtained from the average reward CMDP literature.

2. It is not clear from the related works if the model-free best policy identification (BPI) approach has been considered in the literature solely for unconstrained MDP. If yes, then the best regret bound in that category should be pointed out.

3. The term $M$ in Lemma 2 has not been explicitly defined.

4. There should be a policy initialization in Algorithm 1.

5. Please use a different notation for the coefficients in $(11)$. The current one is similar to the notation of an action.

6. It seems that the number of optimization variables in $(11)$ is $\mathcal{O}(A^{SH})$ which could make the problem prohibitive for a large state space. This should be clearly stated in the paper.

7. Assumption 1 indicates that the design of the algorithm requires knowledge about the optimal occupancy measure which is highly unlikely in practice.

8. It seems that Assumption 2 is redundant for finite state and action spaces.

9. The policy identification stage is run for $\mathcal{O}(MKH)$ number of steps and as stated before, $M$, in the worst-case can be $\mathcal{O}(A^{SH})$. Why this bound does not appear in the final regret should be intuitively explained.

10. Theorem 3 dictates the BPI result assuming perfect pruning which does not happen with at most $\mathcal{O}(K^{-0.1})$ probability. In the introduction, this probability is mentioned to be $\mathcal{O}(1/\sqrt{K})$. Please clarify.

---

> ### Author Response · Authors · 2023-11-15
>
> Response:
>
> We thank the reviewer's positive comments. Our responses to your answers can be found below.
>
> **Q1.** Thanks for the suggestion! We will add the discussion on its connection to average reward CMDPs.
>
> **Q2.** Most BPI algorithms for MDPs are model-based. Exisitng model-free BPI algorithms output a mixed policy that chooses one policy from all $K$ used policy uniformly at random instead of a single stochastic policy. We will add a discussion to the related work.
>
> **Q3.** $M$ is the same as the $M_q$ defined on page 4. We will clarify this in the revision.
>
> **Q4.** Sorry for the confusion. The policy initialization is the same as that of Triple-Q. We will clarify this in the revision.
>
> **Q5.** Good point! We will change it to $\alpha_m$ in the revision.
>
> **Q6.** The number of optimization variables is $M,$ the number of greedy policies after the pruning phase. $M$ is upper bounded by $2^N,$ where $N$ is the number of constraints. We will clarify this in the revision.
>
> **Q7.** In practice, we choose small $\epsilon$ and $\epsilon'$ to use. When the assumption does not hold, the algorithm cannot provide the performance guarantees in the paper.
>
> **Q8.** Can you please elaborate? It is not clear to us if the assumption automatically holds for finite state and action spaces.
>
> **Q9.** $M$ is bounded by $2^N,$ where $N$ is the number of constraints. Also, $\sum_{i=1}^M a_i = 1$ so the policy identification stage includes $K$ episodes.
>
> **Q10.** Sorry for the typo. It should be $O(K^{-0.1})$ in the introduction as well.

---

> ### Author Response · Authors · 2023-11-20
> **Any further questions/comments**
>
> We have uploaded a revision that incorporates your comments. The author-reviewer discussion ends in a few days, we would like to know whether you have any additional questions/comments we can address.

---

### Official Review · Reviewer_QLDH · 2023-11-01

**Soundness:** 3 good
**Presentation:** 4 excellent
**Contribution:** 3 good
**Rating:** 6
**Confidence:** 2

**Summary:**

The paper considers the reinforcement learning problem for constrained MDPs in the tabular setting, and proposes a model-free algorithm that returns a policy with with sublinear $\tilde{\mathcal{O}}(\sqrt{K})$ regret and constraint violation with high probability.

**Strengths:**

- Exploiting specific structural properties of policies and occupancy measures in the constrained MDP case, the paper proposes an effective model-free learning algorithm with a good regret and acceptable constraint violation performance. Instead of best-iterate convergence, a stronger regret result was proved. These results may be good contributions.
- The paper is extremely well-written. The algorithm design and analysis were discussed very clearly.

**Weaknesses:**

- Although the algorithm achieves a better regret bound compared to Triple-Q in (Wei et al., 2022a), this improvement comes at the expense of increased constraint violation. Is there a tradeoff between regret and constraint violation? If so, is it possible to achieve this tradeoff by using different hyperparameters?

**Questions:**

- How does the minimum state-exploration probability $p_{min}$ in Assumption 3 appear in the regret and constraint violation bounds?

- Should Assumption 3 hold for any greedy policy $\pi$? It looks a little strong in its current form.

---

> ### Author Response · Authors · 2023-11-15
>
> Response:
>
> We thank the reviewer's positive comments. Our responses to your answers can be found below.
>
> **Weakness**. This is a great question. There may be a trade-off between the regret and constraint violation. However, the algorithm in the paper cannot provide such a trade off by tuning the hyperparameters. This is a great problem for future research.
>
> **Q1**. In Appendix B, $p_{\min}$ shows up in the lower bound on the occupancy measure (18), and determines the constant associated the probability that stochastic decisions are correctly classified (i.e., probability in (19)).
>
> **Q2**. Yes, it needs to hold for any greedy policy. We later relax this assumption when we add the multi-solution pruning phase.

---

> ### Author Response · Authors · 2023-11-20
> **Any further questions/comments**
>
> We have uploaded a revision that incorporates your comments. The author-reviewer discussion ends in a few days, we would like to know whether you have any additional questions/comments we can address.

---

> > ### Comment · Reviewer_QLDH · 2023-11-22
> >
> > I would like to thank the authors for their clarification. I will keep my score as it is. Under similar assumptions as Triple-Q as in (Wei et al., 2022a), the regret can be improved, but this comes at the expense of an increased constraint violation. I think the contribution could be much stronger if the tradeoff between these two extremes could be achieved.

---

### Official Review · Reviewer_3osr · 2023-11-05

**Soundness:** 3 good
**Presentation:** 3 good
**Contribution:** 3 good
**Rating:** 6
**Confidence:** 3

**Summary:**

This paper provides a model-free algorithm for online constrained MDP. The algorithm is based on Triple Q and a novel pruning-refinement-identification algorithm. This paper also highlights a limited stochasticity property of the optimal policy for constrained MDP that has been overlooked in the literature. The proposed algorithm enjoys both sublinear regret and constraint violation, which improves the existing algorithm in the literature. Simulation results also show performance improvement.

**Strengths:**

1. The paper is well-written.
2.

**Weaknesses:**

My only suggestion for improving the clarity is to add a short review of Triple-Q to make the paper more self-contained. Other questions are discussed in the next box.

**Questions:**

Q1: in equation (3), does $\rho^n$ mean the exponential of $\rho$, or just a constant that differs with $n$? If it is an exponential of $\rho$, can the authors explain why considering this special form? What's the difficulty of considering general $\rho_n$? If it is indeed a constant that takes different values with $n$, then I suggest using $\rho_n$ to avoid confusion.

Q2: In Algorithm 1, what is K is unknown? How to implement the algorithm? Does Theorem 1-3 still hold?

Q3: In section 7, why does Triple Q have a much smaller negative constraint violation? Is it because Triple Q becomes very conservative in the end? But shouldn't the conservativeness reduce as learning continues? Besides, instead of total constraint violation, what's the total number of episodes or stages of constraint violation?

---

> ### Author Response · Authors · 2023-11-15
>
> Response:
>
> We thank the reviewer's positive comments. Our responses to your answers can be found below..
>
> **Weakness**. Thank you very much for the suggestion! We will add the review of Triple-Q in the revision.
>
> **Q1**. It is a constant a constant for constraint $n,$ not the exponential. We will use $(n)$ in the revision to avoid this confusion. This paper the subscripts are related to time (step index, episode index, etc) so we feel it is better to superscript as the constraint index.
>
> **Q2**. When $K$ is unknown, we may use the standard doubling trick, i.e., running the algorithm by increasing the assumed horizon from $T$, $2T,$ to $4T,$ $\cdots$ until the process ends.
>
> **Q3**. Yes, the reviewer is correct that Triple-Q is conservative and aims at achieving zero constraint violation, so it has smaller negative constraint violation. While the conservativeness reduces under Triple-Q, the plot shows the total constraint violation so it continues to decreases. We did not count the number of episodes where the constraint is violated because the constraint violation is defined on the utility-value function, which is the expected total utility, averaged over many episodes.

---

> ### Author Response · Authors · 2023-11-20
> **Any further questions/comments**
>
> We have uploaded a revision that incorporates your comments. The author-reviewer discussion ends in a few days, we would like to know whether you have any additional questions/comments we can address.

---

> > ### Comment · Reviewer_3osr · 2023-11-22
> >
> > Thank you for the clarification! I don't have more questions.

---

### Official Review · Reviewer_ost9 · 2023-11-06

**Soundness:** 2 fair
**Presentation:** 2 fair
**Contribution:** 2 fair
**Rating:** 5
**Confidence:** 4

**Summary:**

In the paper, the authors propose a model-free algorithm for deterministic CMDP (deterministic, in terms of rewards and constraints) which guarantees \sqrt{T} regret bound and constraint violation. Moreover, the algorithm proposed outputs a near-optimal policy with high probability.

**Strengths:**

The paper is well written. Furthermore, the authors propose the first model-free algorithm achieving \sqrt{T} regret and violations in CMDPs which outputs a near optimal policy, which is a non-trivial result.

**Weaknesses:**

1) Since the paper refers to deterministic CMDP (and even assuming the generalisation to stochastic rewards and constraints to be trivial), the notion of violation proposed seems to be weak. Indeed, [Efroni et al., 2020]  model-based methods, achieves optimal sublinear violation when the cancellations between episodes are not possible.
2) The algorithm strongly relies on the Triple-Q algorithm, employing it as subroutine. Thus, the algorithmic novelty is partial.
3) The assumption that the CMDP’s LP has a unique solution is strong. Since it is relaxed in the second part of the paper, I do not see any reason to focus half of the paper on this case. Same reasoning holds for assumption 3.
4) The theoretical results hold only for Large K, while no guarantees are provided if the number of episodes is small.
5) In chapter 6, when the assumption on unique solution is relaxed, the authors introduce additional strong assumptions. For example, Assumption 4 states that the algorithm is given as input a lower bound on the probability of visiting every state-action pairs (when it is not 0) under the optimal policies belonging to the extreme point of the decision space.
6) Given that the main novelty of the paper concerns the model-free nature of the algorithm (indeed, model-based algorithms achieves better theoretical guarantees), the authors should devote more space clarifying which is the improvement in terms of the computational complexity of the algorithm proposed with respect to prior works.

**Questions:**

Triple-Q assumes that salter condition holds. I assume this must be true even in your case, since Triple-Q is employed in PRI algorithm. Is it right?

---

> ### Author Response · Authors · 2023-11-15
>
> Response:
>
> We sincerely thank the reviewer for the great comments and questions. Please see our detailed responses below. We believe we have addressed/clarified your concerns/questions. Please let us know if you have any further comments. We would also greatly appreciate if you could reevaluate the rating based on our response. Thanks!
>
> **Weakness 1**: We thank the reviewer for this comment. While the constraint violation defined in the paper allows the cancellation across episodes, we can guarantee $O(\sqrt{K}\log K)$ violation when the cancellation is not allowed by making some minor modification to the algorithm. In particular, in the policy refinement and identification phases, instead of using the $M$ greedy policy in a round-robin fashion, we can use a mixed policy that chooses policy $m$ with probability $a_m.$ With that modification, we will have
> * The Pruning phase consists at most $\sqrt{K}$ episodes, so resulting in at most $O(\sqrt{K})$ violation.
> * During the refinement phase, based on the result on page 16 of the supplemental material (Inequality (27)), we have with a high probability that the mixed policy used in the $t$th iteration has a regret bounded by $O(\frac{\log K}{(t-1)\sqrt{K}})$ for $t\geq 2.$ The same result holds for constraint violation, i.e., with a high probability, the constraint violation of the policy used in the $t$th iteration is bounded by  $O(\frac{\log K}{(t-1)\sqrt{K}}).$ The $t$th iteration includes $\sqrt{K}$ episodes and the refinement phase includes $\sqrt{K}$ iterations. Therefore, with a high probability, the total violation without canceling across episodes is $O(\sqrt{K}\log K).$
> * The mixed policy used in the policy identification phase has a constraint violation bound of $O(\frac{\log K}{\sqrt{K}}).$ The policy is used for $K$ episodes, so the violation is bounded by   $O(\sqrt{K}{\log K}).$
>
> Therefore, the total violation is bounded by $\tilde{O}(\sqrt{K})$ by using definition (5) in [Efroni et al., 2020]. We will add this in the revision.
>
> **Weakness 2**. Our results hold if we replace Triple-Q with any model-free algorithm for CMDPs with sublinear regret and sublinear constraint violation. PRI can be viewed as a ``meta-algorithm'' that builds on any model-free CMDP algorithm with sublinear regret and constraint violation to achieve  $\tilde{O}(\sqrt{K})$ regret and constraint violation and to output a single stochastic policy.
>
> **Weakness 3**. We present the special case phase first for readability. Focusing on the special case where the problem has a unique optimal solution allows us to explain the key ideas behind the algorithm and makes the analysis easier to follow. We can directly present the general case as the reviewer suggested, but it will not save much space because the results we established for the single-solution case are used for the multi-solution case. Therefore, there is minimal repetition in the current structure.
>
> **Weakness 4**. The reviewer is correct that the result holds only for large $K.$ It is common that regret bounds in bandits and reinforcement learning hold only for large $K.$ Note that order-wise results such as $O(\sqrt{K})$ means it holds when $K$ is sufficiently. Big O notation defines the limiting behavior, i.e. when $K$ goes to infinity.
>
> **Weakness 5**.  Assumption 4 is similar to Assumption 1. It states that if a state-action pair is visited under an optimal policy, then the visiting probability is lower bounded. We believe it is a reasonable assumption and easy to satisfy for problems with finite state and action spaces.
>
> **Weakness 6**. Thanks for the suggestion! Model-free algorithms require a memory complexity of $O(HSA)$ for maintaining the Q-Table  while model-based algorithms require a memory complexity of $O(HS^2A)$ for maintaining all transition probabilities. This is the main advantage of using model-free algorithms in practice. We will clarify this in the revision.
>
> **Question about Slater's condition**. Yes, the reviewer is correct. We will clarify this in the revision.

---

> > ### Author Response · Authors · 2023-11-19
> > **Any further questions/comments**
> >
> > We want to follow up to see whether our response addresses your concerns. Please don't hesitate to let us know if you have any other questions/comments. Thanks!

---

> ### Author Response · Authors · 2023-11-20
>
> We have uploaded a revision that incorporates your comments. The author-reviewer discussion ends in a few days, we would like to know whether your concerns have been addressed and whether you have any additional questions/comments we can address. We again would appreciate if you could reconsider your rating based on our response.

---

### Comment · Area_Chair_TKMY · 2023-11-20

Given that previous works have demonstrated zero constraint violation using Slater's condition, a condition also employed in this paper, can similar guarantees be established in the context of this study? It is unclear why this cannot be done, since the conservative constraints can lead to zero constraint violation, and typically the impact on the objective is the same order; thus seems like the paper could be extended following existing works in the area to obtain zero constraint violation. More explanations and insights would help related to this aspect.

Also, with the modification, when we have probability K^{-0.1} of possibly having large violation, does that not take any advantage out of the picture. Overall regret is bounded by K(1/\sqrt{K} + 1/K^{0.1}) and thus the gap is K^{9/10}. Seems like the bad event is dominating here. Am I missing something? The results become even worse with multiple optimal solutions, which is the general case.

---

> ### Author Response · Authors · 2023-11-21
>
> Both are great questions!
>
> For zero constraint violation, Triple-Q achieves zero violation because the regret is $O(K^{0.8})$, which is higher than $O(K^{0.5})$ regret. The design of the algorithm reduces the constraint violation with a conservative design, which increases the regret but the increase is smaller than $O(K^{0.8})$, so it does not affect the regret in the analysis of Triple-Q. In our paper, the algorithm achieves $O(K^{0.5})$. We tried the conservative design, which, however, increases the regret and makes it higher than $O(K^{0.5})$. By now, we are not sure whether we can use a conservative constraint without increasing the order of regret but we continue working on it.
>
> About your second question, our result is a high probability result, not with probability one. In other words, the regret and constraint violation are both $O(K^{0.5})$ with a high probability $1-O(K^{-0.1})$. This is because we used the Markov inequality to show that the probability that all stochastic decisions are correctly identified with probability $1-O(K^{-0.1})$ (Theorem 1). Conditioned on this event, the regret and constraint violation in the refinement and identification phases are $O(K^{0.5})$ with probability $1-O(K^{-0.5}).$ We would like to comment (1) while the high probability result is not as strong as the probability one result like the results in Triple-Q, this type of result is not uncommon in the literature. (2) We believe this result is not tight and is a result of the Markov inequality. We are currently working on using better concentration result to improve the result.

---

> > ### Comment · Area_Chair_TKMY · 2023-11-22
> >
> > This seems to confirm my concern that the expected regret is worse than the state of the art in its current form, correct? If so, then the comparison with the state of the art is incorrect and the expected regret is significantly higher than the state of the art.

---

> ### Author Response · Authors · 2023-11-22
> **An easy fix**
>
> Dear Area Chair,
>
> Thanks again for this very insightful question! It turns out that there is an easy fix. Instead of running Triple-Q just once at the pruning phase, we can modify the algorithm and run Triple-Q $\log K$ times, independently. We then classify each decision, greedy or stochastic, based on the majority rule. Since each decision (greedy or stochastic) is correctly classified with probability $1-\tilde{O}(K^{-0.1})$ under each run of the Triple-Q, the probability a decision is correctly classified with the majority rule is $1-\tilde{O}(e^{-(\sqrt{K}\log K)/16})$ based on the Chernoff bound, or at least $1-O(\frac{1}{K^2})$ i.e., Theorem 1 now holds with probability $1-O(\frac{1}{K^2}).$ So the large violation occurs with $O(\frac{1}{K^2})$ and its contribution to regret is $O(1/K).$ We also note that Theorem 2 holds with probability $1-O(1/\sqrt{K})$ and Theorem 3 holds with probability  $1-O(1/K)$ so the large violation events under Theorem 2 and 3 also do not change the order of the regret. The same idea can be used to the multi-solution case to improve the probability from $1-O(K^{-0.02})$ to $1-O(K^{-2}).$ In that case the probability of bad events can be bounded by $\tilde{O}(K^{-0.5})$, generated from phase 2 and phase 3. With this easy fix, we now can guarantee $\tilde{O}(\sqrt{K})$ regret and constraint violation with probability one as in the Triple-Q paper.
>
> Additionally, we found that zero constraint violation is accomplishable. With a conservative constraint $\rho + \epsilon_\rho$, where $\epsilon_\rho = \frac{2H\sqrt{K} + 4\frac{H}{\sqrt{\epsilon'}}\sqrt{2K\log K} + H^2SAK^{0.25}}{2K+\sqrt{K}logK}$, the violation will be substracted at least $(2K + \sqrt{K}logK) \epsilon_\rho$, which ensures the zero constraint violation based on cancellation. On the other hand, with Slater's condition, we can conclude that $V^* - V^{*,\epsilon_\rho} \leq \frac{H\epsilon_\rho}{\delta}$ ,where $\delta$ is defined in Slater's condition. In that case the additional terms on regret will also be of $\tilde{O}(\sqrt{K})$.
>
> We sincerely thank the area chair for raising this great question, which helped us strengthen the paper!

---

### Meta-Review · Area_Chair_TKMY · 2023-12-05

**Metareview:**

This paper considers the best policy identification problem in online Constrained Markov Decision Processes (CMDPs). We are interested in algorithms that are model-free, have low regret, and identify an optimal policy with a high probability.

Strengths:
The paper claims to achieve the state of the art results for CMDP. The results, if correct, improve the state of the art significantly, and are a significant improvement in the area. The proof technique has novel components.

Weaknesses:
The paper claim in the first submitted version had issues, as pointed by Reviewer S5ah, where Theorem 3 had the high probability event with probability 1-O(K^{-.1}), while Introduction had claim of 1-O(K^{-.5}). This discrepancy suggested an inconsistency in the reported regret, creating an impression of \sqrt{K} regret even for the expected regret based on Introduction. The authors acknowledged an error in the Introduction, diminishing the expected regret result compared to the state of the art. In discussions with the AC, the authors proposed an approach to enhance the result, achieving O(\sqrt{K}) expected regret with zero constraint violation. However, the reviewers were unable to verify this improvement without it being presented in the complete manuscript.

While both the reviewers and the AC anticipate that the paper could be substantially strengthened with these proposed changes, the lack of verification due to the incomplete presentation raises concerns. Consequently, it is recommended that the authors incorporate these adjustments, and believe that the paper with its strong results would be publishable at any top-tier ML venue.

**Justification For Why Not Higher Score:**

The paper, as in the submitted pdf, do not improve the state of the art results. The proposed modification by the authors needs complete verification based on the revision of the paper.

**Justification For Why Not Lower Score:**

N/A

---

### Decision · Program_Chairs · 2024-01-16

Reject